# When Human Data Runs Out: Self-Supervised Reasoning via Negotiation Self-Play

## Abstract

The reasoning ability of large language models (LLMs) has canonically relied on costly expert-labeled data, a resource now nearing depletion. Existing alternatives, such as Chain-of-Thought prompting or multi-agent debate, either suffer from prompt sensitivity or assume binary correctness, limiting their applicability to open-ended reasoning tasks. This raises a fundamental challenge: *how can we construct scalable supervision that drives diverse yet stable reasoning without external labels?* Verbal interaction offers the most natural source of new supervision signals; among the tasks that feature such interaction between AI agents, negotiation stands out as particularly suited for reasoning enhancement. We introduce Language model Self-play via Scorable negotiation Game (LSSG), a paradigm that frames reasoning enhancement as a two-player negotiation game with continuous, outcome-based rewards. Unlike prior numerical- or annotation-based games, our formulation pioneers negotiation in the language space, providing dense, interpretable signals for stable optimization at scale. LSSG combines *behavioral cloning* from real dialogues with *self-play refinement* that balances diversity and stability, yielding sustainable reasoning improvement. Across seven benchmarks, including WinoGrande, CSQA, CB, SST2, LogiQA2, MedMCQA, and CMMLU, LSSG consistently outperforms strong baselines. These results demonstrate LSSG as a scalable and robust paradigm for long-term reasoning self-supervision in LLMs.

## 1 Introduction

Most existing methods for enhancing the reasoning ability of large language models (LLMs) rely on labeled data (Morishita et al., 2024; Peng et al., 2025; Zhang et al., 2024), but acquiring such data is costly and requires domain expertise. In fact, large-scale, freely accessible human-labeled datasets have already been substantially depleted (Silver & Sutton, 2025; Villalobos et al., 2024; Chen et al., 2024), further highlighting the urgency of developing methods that can *continuously improve reasoning capabilities without reliance on expert-labeled data*.

A straightforward method to enhance reasoning is prompt engineering, where models are guided to think step-by-step via Chain-of-Thought (CoT) prompting (Wei et al., 2022; Chen et al., 2024). While effective, CoT depends heavily on careful prompt design and suffers from sensitivity to prompt wording and ordering. Another promising direction is to use self-evaluation or multi-agent debate, where either two LLMs debate and a judge model selects the better response (Khan et al., 2024; Estornell & Liu, 2024; Sternlicht et al., 2025; Eo et al., 2025), or one LLM critiques and revises its own answers (Jacob et al., 2024; Miao et al., 2024; Besta et al., 2024; Kawabata & Sugawara, 2024). These methods are motivated by the observation that verifying correctness can be easier than generating correct answers. Nevertheless, debate protocols often assume binary correctness, making them difficult to generalize to more open-ended reasoning tasks where answers may be partially correct, subjective, or multi-faceted. These limitations motivate a deeper examination of what fundamentally hinders reasoning enhancement without supervision.

Enhancing the reasoning ability of LLMs without access to labeled data presents two major challenges. The first challenge is the lack of explicit supervision. Without labeled data, models cannot be refined through standard supervised learning, often resulting in unstable or drifting intermediate reasoning steps. The second challenge is limited generalization and robustness. In the absence of

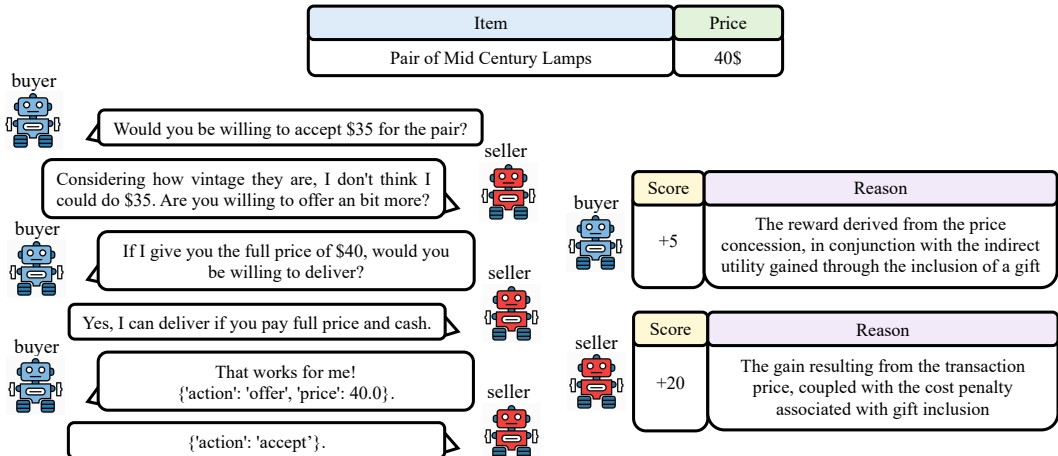

Figure 1: An example of the Scorable Negotiation Game (SG), where two agents negotiate over the item 'Pair of Mid Century Lamps'. The left side shows the dialogue between the buyer and seller agents, while the right side presents the scores they receive along with the reasons for each score.

human-provided feedback, models tend to memorize surface patterns rather than acquire genuine reasoning strategies, which degrades their out-of-distribution performance and makes them more susceptible to noisy or adversarial inputs.

To address these challenges, we propose Language model Self-play via Scorable Negotiation Game (LSSG). LSSG formulates reasoning as a two-player *scorable negotiation game*, thereby generating automatic supervision signals through self-play to compensate for the lack of labeled data. Furthermore, by incorporating diverse negotiation contexts and adversarial strategies, LSSG exposes and corrects reasoning weaknesses, ultimately improving generalization and stability across unseen tasks.

**Why a new negotiation game.** Prior work has explored negotiation and bargaining games as testbeds for multi-agent communication and strategy learning (Lewis et al., 2017; Deng et al., 2024). These settings, however, are typically designed with domain-specific utilities (e.g., discrete item division, win/lose outcomes) and often require additional human annotations or learned reward models to determine success. Crucially, most of these games are numerical toy settings or non-linguistic simulations, which restrict their ability to capture reasoning behaviors expressed through natural language. Our work makes a conceptual shift: rather than operating in simplified numeric environments, we introduce a negotiation game that unfolds entirely in the language space, enabling reasoning self-play to be trained and evaluated directly through natural dialogues. This shift removes the reliance on handcrafted utilities or binary outcomes, providing continuous and interpretable supervision signals. In contrast, we define a *scorable negotiation game* with continuous, outcome-based rewards that directly reflect price concessions and gift trade-offs. This formulation provides dense, differentiable learning signals rather than binary judgments, enabling stable policy optimization through self-play.

LSSG consists of two schemas: *generalization-aware behavioral cloning* and *stability-aware self-play*. For *generalization-aware behavioral cloning*, LSSG pretrains buyer and seller policies via behavioral cloning on real negotiation dialogues. To prevent overfitting to frequent action patterns, LSSG introduces entropy maximization to encourage a smoother action distribution and adds KL regularization to retain language plausibility. This combination improves generalization across diverse negotiation scenarios. For *stability-aware self-play*, LSSG fine-tunes both policies via self-play with advantage-weighted updates and entropy regularization to stabilize policy improvement. To further enhance robustness, LSSG incorporate semantic diversity loss (encouraging multiple valid reasoning paths via sentence embeddings) and emotional stability loss (regularizing sentiment distribution toward neutrality). Applied across seven question-answering benchmarks including Wino-Grande (Sakaguchi et al., 2021), CSQA (Talmor et al., 2019), CB (De Marneffe et al., 2019), SST2 (Socher et al., 2013), LogiQA2 (Liu et al., 2023), MedMCQA (Pal et al., 2022), and CMMLU

(Li et al., 2024), LSSG consistently enhances the reasoning capability of LLMs, demonstrating its effectiveness and scalability across diverse reasoning tasks.

## 2 SCORABLE NEGOTIATION GAME

Consider a Rubinstein's alternating-offer bargaining model with cheap talk. The game is defined as a tuple $\mathcal{G} = \langle \mathcal{S}, \mathcal{A}^B, \mathcal{A}^S, \mathcal{T}, \mathcal{R}^B, \mathcal{R}^S, \gamma \rangle$, where $\mathcal{S}$ is the state space, $\mathcal{A}^B$ and $\mathcal{A}^S$ are the action spaces of a buyer $B$ and a seller $S$, $\mathcal{T}$ is the transition function of states, $\mathcal{R}^B$ and $\mathcal{R}^S$ are utility functions, and $\gamma \in [0, 1]$ is the discount factor. At the start, the buyer $B$ observes information about an item from a seller $S$, including the name and price. Then the players negotiate over the trade of this item, where each can choose to make a new offer, accept or reject the current offer, or simply engage in casual communication (Figure 1). Both players are self-interested and rational, aiming to maximize their own utility: the seller $S$ seeks to keep the price near the original, possibly by adding small incentives (e.g. giving gifts or free shipping), while the buyer $B$ aims to lower the price or obtain additional bonus. Gifts and other conveniences provided by the seller, together with the price, are all factored into both parties' utility. An offer is accepted when either side replies with "accept", ending the game with a joint win. Both players can choose to "quit" at any time, ending the game with both getting nothing. The players interact in alternating turns, and the game is limited to 15 turns.

The components of the tuple are formally specified as follows: **(1)** The action space is $\mathcal{A}^i = \{\texttt{offer}(P), \texttt{accept}, \texttt{quit}\} \times \Sigma$, for each player $i \in \{B, S\}$, where $P \in \mathbb{R}^+$ is a proposed price which already incorporates the effects of any additional incentives offered by the seller, and $\sigma \in \Sigma$ represents a free-form message for negotiation. At each timestep, a player either proposes a price, accepts the last proposed price, or quits the game, while the message component is optional. **(2)** The state space $\mathcal{S}$ is defined as $s_t = (t, i_t, \tau_t)$, which includes (i) the current game step $t \in \{1, \ldots, T\}$, (ii) the active agent index $i$, and (iii) the trajectory $\tau_t = ((a_0, \ldots, a_{t-1}), (\sigma_0, \ldots, \sigma_{t-1}))$. **(3)** The transition function is given by $\mathcal{T} : \mathcal{S} \times \mathcal{A}^B \times \mathcal{A}^S \to \Delta(\mathcal{S})$, where $\Delta(\mathcal{S})$ denotes the probability simplex over $\mathcal{S}$. If either player chooses accept, the game transitions to a terminal state; if both continue negotiation, $t \leftarrow t + 1$, $i_t \leftarrow 1 - i_t$, and the state is updated with the latest price and messages; and if $t = T$ with no agreement reached, the game transitions to a failure state. **(4)** The utility functions $\mathcal{R}^B$ and $\mathcal{R}^S$ are defined using a concession ratio that measures the extent of compromise relative to the initial offer. For the buyer, with initial price $P_0$ (the seller's starting price) and final transaction price $P_d$, the concession ratio is $r = 1 - P_d/P_0$, yielding the reward $\mathcal{R}^B = \alpha_1 \cdot (1 - P_d/P_0)$. For the seller, a higher transaction price yields higher reward, but additional incentives (e.g. gifts) incur costs, leading to $\mathcal{R}^S = \alpha_2 \cdot P_d/P_0$. Here, $\alpha_1$ and $\alpha_2$ are hyperparameters that balance price concession, gift utility, and cost. **(5)** Finally, the objective is that each agent $i \in \{B, S\}$ learns a policy $\pi^i : \mathcal{S} \to \mathcal{A}^i$ to maximize the expected cumulative reward $\max_{\pi^i} \mathbb{E}_{\pi^i, \pi^j} \left[ \sum_{t=1}^T \gamma^t R^i \cdot \mathbf{1}_{\text{accept}} \right]$, where $\mathbf{1}_{\text{accept}}$ is an indicator function equal to 1 if an offer is accepted and 0 otherwise.

## 3 LANGUAGE MODEL SELF-PLAY VIA SCORABLE NEGOTIATION GAME

### 3.1 GENERALIZATION-AWARE BEHAVIORAL CLONING

Behavioral cloning enables agents to approximate the optimal negotiation policy from real buyer-seller dialogues. Formally, let $\mathcal{D} = \{(s_t, a_t)\}_{t=1}^T$ denote a dialogue trajectory of length $T$. The generalization-aware behavioral cloning (GABC) objective for the buyer is defined as

$$\mathcal{L}_{\text{GABC}}^{\text{B}}(\pi^B) = -\frac{1}{T} \sum_{t=1}^T \log \pi^B(a_t^B \mid s_t) - \beta_1 \cdot \frac{1}{T} \sum_{t=1}^T \underbrace{\left[ -\sum_{a \in \mathcal{A}} \pi^B(a \mid s_t) \log \pi^B(a \mid s_t) \right]}_{\text{Entropy } H(\pi^B(\cdot|s_t))}$$

$$+ \beta_2 \cdot \underbrace{\sum_{a \in \mathcal{A}} \pi^B(a|s_t) \log \frac{\pi^B(a|s_t)}{\pi_0^B(a|s_t)}}_{\text{KL}[\pi^B \| \pi_0^B]}. \tag{1}$$

Similarly, for the seller policy $\pi^S$ we have

$$\mathcal{L}_{\mathrm{GABC}}^{\mathrm{S}}(\pi^S) = -\frac{1}{T}\sum_{t=1}^{T}\log \pi^S(a_t^S \mid s_t) - \beta_3 \cdot \frac{1}{T}\sum_{t=1}^{T} H\big(\pi^S(\cdot \mid s_t)\big) + \beta_4 \cdot \mathrm{KL}[\pi^S \,\|\, \pi_0^S]. \quad (2)$$

The joint loss $\mathcal{L}_{\mathrm{GABC}}$ of generalization-aware behavioral cloning is the weighted sum

$$\mathcal{L}_{\mathrm{GABC}} = \frac{1}{2}\mathcal{L}_{\mathrm{GABC}}^{\mathrm{B}}(\pi^B) + \frac{1}{2}\mathcal{L}_{\mathrm{GABC}}^{\mathrm{S}}(\pi^S),$$

where the first term in each loss is the negative log-likelihood that clones expert actions, the second term maximizes policy entropy to encourage diverse actions, and the last KL term constrains updated policies to stay close to the reference distributions $\pi_0^B$ and $\pi_0^S$, thereby preserving language fluency while allowing controlled exploration. Here, $\pi_0^B$ and $\pi_0^S$ denote frozen reference (pre-trained) buyer and seller policies, and $\beta_1, \beta_2, \beta_3, \beta_4$ are hyperparameters weighting entropy regularization and KL penalties.

## 3.2 STABILITY-AWARE SELF-PLAY

Following the generalization-aware behavioral cloning phase, the agents acquire a preliminary ability to engage in the Scorable Negotiation Game. To further enhance their performance, we introduce a stability-aware self-play stage. Self-play has been widely demonstrated to allow agents to iteratively bootstrap their capabilities, progressively surpassing the limitations imposed by fixed human demonstrations (Silver et al., 2017). Let the joint policy be $\Pi = \pi^B \times \pi^S$ and define a stochastic trajectory $\tau = \{(s_t, a_t^B, a_t^S, r_t^B, r_t^S)\}_{t=1}^{T} \sim \Pi$, where $s_t$ is the state at step $t$, $a_t^i$ the action of agent $i \in \{B, S\}$, and $r_t^i$ the immediate reward. We seek to maximize the expected return

$$\mathcal{J}(\pi^B, \pi^S) = \mathbb{E}_{\tau \sim \Pi}\Big[\sum_{t=1}^{T}\gamma_t(r_t^B + r_t^S)\Big],$$

where $\gamma$ is a discount factor. For the buyer, the gradient of $\mathcal{J}$ w.r.t. parameters $\theta_B$ under the policy gradient theorem is

$$\nabla_{\pi^B}\mathcal{J} = \mathbb{E}_{\tau \sim \Pi}\left[\sum_{t=1}^{T}\hat{A}_t^B \nabla_{\pi^B}\log \pi^B(a_t^B \mid s_t)\right], \quad (3)$$

where $a_t^B = Q_B(s_t, a_t^B) - b(s_t)$ is the advantage function estimating the relative value of action $a_t^B$ over a baseline $b(s_t)$. An analogous formulation holds for the seller policy $\pi^S$.

To improve update stability, we adopt a GRPO-style advantage-weighted likelihood maximization with entropy regularization (Shao et al., 2024), leading to the *stability-aware self-play* (SASP) objectives of agent buyer $B$:

$$\mathcal{L}_{\mathrm{SASP}}^{\mathrm{B}}(\pi^B) = -\mathbb{E}_{\tau \in \mathcal{D}^B}\left[\sum_{t=1}^{T}\hat{A}_t^B \log \pi^B(a_t^B \mid s_t) - \lambda_1 H(\pi^B(\cdot \mid s_t))\right], \quad (4)$$

where $\hat{A}_t^i$ are Monte Carlo advantage estimates, $\mathcal{D}^B$ denotes the subset of successful trajectories used for policy update of agent buyer $B$, $\lambda_1$ controls the strength of entropy regularization for the buyer policy, encouraging exploration by preventing the action distribution from collapsing too early. Similarly, for the agent seller $S$ we have

$$\mathcal{L}_{\mathrm{SASP}}^{\mathrm{S}}(\pi^S) = -\mathbb{E}_{\tau \in \mathcal{D}^S}\left[\sum_{t=1}^{T}\hat{A}_t^S \log \pi^S(a_t^S \mid s_t) - \lambda_2 H(\pi^S(\cdot \mid s_t))\right]. \quad (5)$$

**Semantic Diversity Regularization.** To avoid degeneration to deterministic outputs, we introduce a semantic diversity loss that penalizes over-similarity to reference responses. Let $\delta(\cdot)$ be a differentiable sentence-BERT embedding function, and define the cosine similarity between generated action $a^t$ and reference $\tilde{a}^t$ as $\cos(\delta(a_t), \delta(\tilde{a}_t))$. The diversity loss is

$$\mathcal{L}_{\mathrm{SemDi}} = \cos(\delta(a_t), \delta(\tilde{a}_t)) - 1,$$

which penalizes overly similar generations, thereby encouraging diversity when minimized jointly with the main loss.

**Emotional Stability Regularization.** Following findings that emotionally polarized outputs can destabilize multi-agent interaction (Bianchi et al., 2024a), we introduce an emotional stability loss:

$$\mathcal{L}_{\text{EmoSt}} = |s(a_t) - \bar{s}|,$$

where $s(a_t) \in [0, 1]$ is the normalized sentiment score of $a_t$ computed using a pre-trained Distil-BERT classifier, and $\bar{s} = 0.5$ represents a target neutral sentiment. This regularization explicitly pulls the distribution of model outputs toward emotional balance.

**Full Objective.** The final objective of language model self-play via the scorable negotiation game (LSSG) integrates both policy-gradient objectives and auxiliary regularizers:

$$
\begin{aligned}
\mathcal{L}_{\text{LSSG}}(\pi^B, \pi^S) &= \alpha_1 \mathcal{L}_{\text{SASP}}^{\text{B}}(\pi^B) + \alpha_2 \mathcal{L}_{\text{SASP}}^{\text{S}}(\pi^S) + \alpha_3 \mathcal{L}_{\text{SemDi}} + \alpha_4 \mathcal{L}_{\text{EmoSt}} \\
&= -\alpha_1 \mathbb{E}_{\tau \in \mathcal{D}^B} \left[ \sum_{t=1}^{T} \hat{A}_t^B \log \pi^B(a_t^B \mid s_t) - \lambda_1 H(\pi^B(\cdot \mid s_t)) \right] \\
&\quad - \alpha_2 \mathbb{E}_{\tau \in \mathcal{D}^S} \left[ \sum_{t=1}^{T} \hat{A}_t^S \log \pi^S(a_t^S \mid s_t) - \lambda_2 H(\pi^S(\cdot \mid s_t)) \right] \\
&\quad + \alpha_3 \left[ \cos(\delta(a_t), \delta(\tilde{a}_t)) - 1 \right] + \alpha_4 |s(a_t) - \bar{s}|,
\end{aligned}
\tag{6}
$$

where $\alpha_1$ and $\alpha_2$ control the relative weight of the buyer and seller SASP losses, $\alpha_3$ and $\alpha_4$ are tunable coefficients controlling the strength of semantic and emotional regularization terms. This formulation jointly optimizes negotiation performance, exploration, semantic diversity, and emotional balance in a unified differentiable framework.

## 4 EXPERIMENTS

To evaluate the effectiveness of the proposed LSSG method, we adopt LLaMA2-7B and LLaMA2-13B (Touvron et al., 2023) as backbone models and conduct training on these architectures. In addition, we include two language games, Adversarial Taboo (Cheng et al., 2024) and the Negotiation Game (He et al., 2018), to broaden the evaluation setting.

### 4.1 EXPERIMENTAL SETUP

**Datasets.** We build upon the CraigslistBargain Dataset (He et al., 2018), which we further expand by crawling 105,267 product records from `sfbay.craigslist.org`, each consisting of a product name and its initial price. Leveraging this enriched dataset, we employ generalization-aware behavioral cloning to train agents for the Scorable Negotiation Game, and subsequently perform stability-aware self-play grounded in these product data. We conduct experiments on seven public available QA datasets: Winogrande (Sakaguchi et al., 2021), CSQA (Talmor et al., 2019), CB (De Marneffe et al., 2019), SST2 (Socher et al., 2013), LogiQA2 (Liu et al., 2023), MedMCQA (Pal et al., 2022), and CMMLU (Li et al., 2024).

**Baselines.** We use LLaMA2-7B and LLaMA2-13B (Touvron et al., 2023) as backbone models and conduct training on these architectures. We consider the following methods:

- **Vanilla Model (V)**: The unmodified backbone models (Touvron et al., 2023) without any additional training or adaptation.

- **Chain-of-Thought (CoT)**: A fine-tuned variant following the chain-of-thought method (Wei et al., 2022), which encourages the model to generate intermediate reasoning steps before producing final answers.

- **Generalization-Aware Behavioral Cloning-Adversarial Taboo Game (G-AG)**: A baseline trained with our generalization-aware behavioral cloning objective on the Adversarial Taboo Game (AG) (Cheng et al., 2024).

- **Generalization-Aware Behavioral Cloning-Negotiation Game (G-NG)**: A baseline trained with the same behavioral cloning strategy on the classic Negotiation Game (NG) (He et al., 2018).

- **Generalization-Aware Behavioral Cloning-Scorable Negotiation Game (G-SG)**: A baseline trained on our proposed Scorable Negotiation Game (SG) using the generalization-aware behavioral cloning framework.

- **Language Model Self-play via Scorable Negotiation Game (LSSG$_i$ ($i = 1, 2, 3$))**: Our two-stage framework Language Model Self-play via Scorable Negotiation Game (LSSG) trained with $i$ iterations of stability-aware self-play reinforcement learning.

**Metrics.** We use accuracy to evaluate the reasoning performance of the model on seven public QA datasets. We evaluate negotiation outcomes using two metrics: win rate and average payoff. Win rate measures the proportion of games where one agent secures strictly more resources than the opponent (ties excluded), while average payoff reflects the overall utility gained per game.

**Implementation details.** Most of our LLM training setups follow Cheng et al. (2024). During the generalization-aware behavioral cloning phase, we set $\beta_1 = \beta_3 = 0.01$, $\beta_2 = \beta_4 = 0.1$, and learning rate as 5e-6. During the stability-aware self-play phase, we set $\lambda_1 = \lambda_2 = 0.2$, $\alpha_1 = \alpha_2 = \alpha_3 = \alpha_4 = 1$, and learning rate as 2e-6. We conduct experiments on a workstation equipped with 8 NVIDIA GeForce RTX 3090 GPUs (24 GB memory each). We evaluate our performance by lm-evaluation-harness (Gao et al., 2024) and NegotiationArena (Bianchi et al., 2024b).

## 4.2 REASONING PERFORMANCE

| Domain | Model | V | CoT | G-AG | G-NG | G-SG | LSSG$_1$ | LSSG$_2$ | LSSG$_3$ |
|--------|-------|-----|-----|------|------|------|-------|-------|-------|
| WinoGrande | LLaMA2-7B | 68.90 | 65.98 | 69.22 | 69.93 | 70.01 | 69.93 | 70.32 | **70.88** |
| | LLaMA2-13B | 72.22 | 69.69 | 71.63 | 72.02 | 72.14 | 72.06 | 72.69 | **72.94** |
| CSQA | LLaMA2-7B | 33.01 | 33.17 | 23.75 | 25.72 | 26.29 | 39.31 | 38.00 | **40.62** |
| | LLaMA2-13B | 46.76 | 46.79 | 42.03 | 45.54 | 45.86 | 59.21 | 62.24 | **62.72** |
| CB | LLaMA2-7B | 42.86 | 42.70 | 33.93 | 41.07 | 41.27 | 50.00 | 48.21 | **51.79** |
| | LLaMA2-13B | 35.71 | 35.82 | 36.25 | 41.35 | 42.86 | 55.06 | 57.42 | **58.93** |
| SST2 | LLaMA2-7B | 49.54 | 49.73 | 49.08 | 55.73 | 56.19 | 63.07 | 66.63 | **70.99** |
| | LLaMA2-13B | 86.04 | 86.22 | 86.47 | 86.93 | 87.04 | 86.21 | 86.24 | **87.17** |
| LogiQA2 | LLaMA2-7B | 25.45 | 26.34 | 25.02 | 25.32 | 25.38 | 27.74 | 28.31 | **29.39** |
| | LLaMA2-13B | 29.77 | 28.18 | 29.31 | 30.31 | 30.47 | 30.85 | 31.42 | **32.76** |
| MedMCQA | LLaMA2-7B | 34.43 | 34.72 | 27.49 | 29.62 | 31.25 | **36.82** | 35.79 | 35.19 |
| | LLaMA2-13B | 38.68 | 38.92 | 31.87 | 37.15 | 37.65 | 40.04 | 40.21 | **40.85** |
| CMMLU | LLaMA2-7B | 27.37 | 27.45 | 27.75 | 29.18 | 29.54 | **31.72** | 31.33 | 30.90 |
| | LLaMA2-13B | 34.78 | 34.93 | 35.04 | 35.41 | 35.93 | 36.51 | 36.81 | **36.97** |

Table 1: Reasoning accuracy (%) of the different methods across multiple tasks. We compute the accuracy on the test set of these benchmarks. **V**: Vanilla LLaMA2 Model, **CoT**: Chain-of-Thought, **G-AG**: Generalization-Aware Behavioral Cloning-Adversarial Taboo Game, **G-NG**: Generalization-Aware Behavioral Cloning-Negotiation Game, **G-SG**: Generalization-Aware Behavioral Cloning-Scorable Negotiation Game, **LSSG$_i$**: Language Model Self-play via Scorable negotiation Game trained with $i$ iterations of stability-aware self-play. Table cell colors are assigned relative to the V baseline, in accordance with the colorbar ▥. -5 -2.5 0 2.5 5

**WinoGrande.** As an extension of the Winograd Schema Challenge, WinoGrande (Sakaguchi et al., 2021) is consisted of sentences with ambiguous pronouns that require commonsense reasoning and world knowledge to resolve. As shown in the first row of Table 1, the LSSG models consistently outperform most of the baselines for both LLaMA2-7B and LLaMA2-13B. In particular, LLaMA2-7B with $LSSG_3$ achieves an accuracy of 70.88%, yielding a +1.98% improvement over the vanilla LLaMA2-7B (68.90%). LLaMA2-13B with $LSSG_3$ achieves an accuracy of 72.94%, yielding a +0.72% improvement over the vanilla LLaMA2-13B (72.22%). In addition, GABC-SG surpass GABC-AG and GABC-NG for both LLaMA2-7B and LLaMA2-13B, which demonstrates that the proposed Scorable Negotiation Game (SG) effectively enhances the reasoning ability of LLMs.

**CSQA.** The CommonsenseQA (CSQA) (Talmor et al., 2019) is a multiple-choice benchmark designed to evaluate a model's ability to perform commonsense reasoning across diverse domains. As reported in the second row of Table 1, our LSSG models yield substantial improvements over the baselines. For LLaMA2-7B, $LSSG_3$ achieves 40.62%, outperforming the vanilla model (33.01%) by +7.61%. Similarly, for LLaMA2-13B, $LSSG_3$ reaches 62.72%, yielding a +15.96% gain over the vanilla LLaMA2-13B (46.76%). These results demonstrate that LSSG is particularly effective on CSQA, where reasoning requires both commonsense knowledge and multi-step inference. Moreover, among the behavioral cloning baselines, G-SG surpasses G-AG and G-NG, further confirming the advantage of our Scorable Negotiation Game (SG) formulation.

**CB.** The CommitmentBank (CB) (De Marneffe et al., 2019) is a natural language inference dataset designed to evaluate whether a hypothesis can be inferred from a premise, focusing on linguistic phenomena such as entailment, contradiction, and neutrality. As shown in the third row of Table 1, our LSSG models significantly improve performance over the baselines. LLaMA2-7B with $LSSG_3$ achieves 51.79%, delivering a +8.93% improvement over the vanilla model (42.86%). For LLaMA2-13B, $LSSG_3$ reaches 58.93%, which corresponds to a remarkable +23.22% gain compared to the vanilla LLaMA2-13B (35.71%). These results indicate that LSSG is highly effective on tasks requiring fine-grained natural language inference. In addition, G-SG outperforms both G-AG and G-NG, further validating the effectiveness of our Scorable Negotiation Game (SG) in enhancing reasoning ability.

**SST2.** The Stanford Sentiment Treebank (SST2) (Socher et al., 2013) is a binary sentiment classification benchmark that evaluates a model's ability to capture fine-grained sentiment in movie reviews. As reported in the fourth row of Table 1, our LSSG models consistently outperform the baselines. With LLaMA2-7B, $LSSG_3$ attains 70.99%, representing a substantial +21.45% improvement over the vanilla LLaMA2-7B (49.54%). With LLaMA2-13B, $LSSG_3$ achieves 87.17%, surpassing the vanilla LLaMA2-13B (86.04%) by +1.13%. These results demonstrate that LSSG significantly boosts smaller models on SST2, while also yielding additional gains on larger ones.

**LogiQA2.** LogiQA2 (Liu et al., 2023) is a multiple-choice benchmark focusing on logical reasoning, requiring models to perform deduction and inference from premises. As presented in the fifth row of Table 1, our LSSG models deliver consistent improvements. With LLaMA2-7B, $LSSG_3$ achieves 29.39%, a +3.94% gain over the vanilla baseline (25.45%). With LLaMA2-13B, $LSSG_3$ reaches 32.76%, improving upon the vanilla LLaMA2-13B (29.77%) by +2.99%. These results confirm that LSSG effectively enhances logical reasoning across both small and large models.

**MedMCQA.** The MedMCQA (Pal et al., 2022) is a large-scale multiple-choice dataset targeting medical domain question answering, which requires domain-specific knowledge and reasoning. As shown in the sixth row of Table 1, LSSG achieves strong improvements over baselines. With LLaMA2-7B, the best result is obtained by $LSSG_1$, which reaches 36.82%, representing a +2.39% gain over the vanilla LLaMA2-7B (34.43%). With LLaMA2-13B, $LSSG_3$ further improves performance to 40.85%, yielding a +2.17% gain over the vanilla LLaMA2-13B (38.68%). These results suggest that LSSG is particularly beneficial in knowledge-intensive tasks like medical QA.

**CMMLU.** The CMMLU (Li et al., 2024) is a comprehensive Chinese multiple-choice benchmark that covers diverse knowledge domains, serving as a challenging test of broad knowledge and reasoning. According to the last row of Table 1, our LSSG models outperform both vanilla and behavioral cloning baselines. With LLaMA2-7B, the best result is achieved by $LSSG_1$, which reaches 31.72%, yielding a +4.35% improvement over the vanilla LLaMA2-7B (27.37%). With LLaMA2-13B, $LSSG_3$ achieves the highest performance with 36.97%, corresponding to a +2.19% gain compared to the vanilla LLaMA2-13B (34.78%).

### 4.3 NEGOTIATION PERFORMANCE

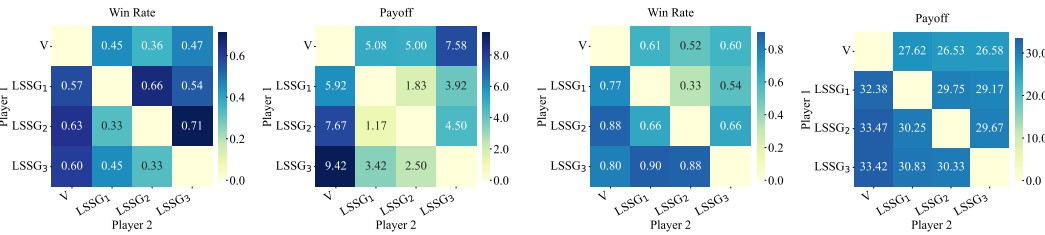

(a) LLaMA2-7B. Seller & Buyer. Win Rate and Pay-off.

(b) LLaMA2-7B. Resource Exchange. Win Rate and Payoff.

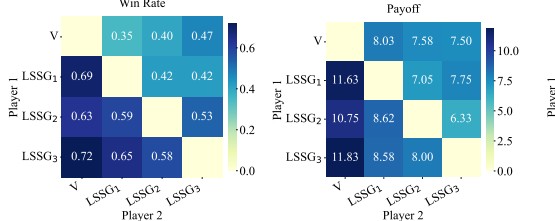
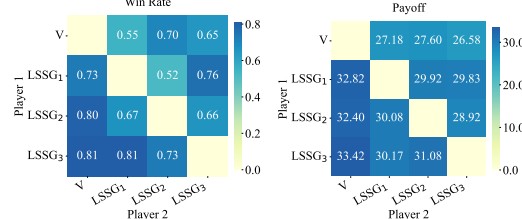

(c) LLaMA2-13B. Seller & Buyer. Win Rate and Pay-off.

(d) LLaMA2-13B. Resource Exchange. Win Rate and Payoff.

Figure 2: Comparison of LSSG models across the two negotiation games.

To verify the negotiation capability of our proposed LSSG, we conduct evaluations on the two negotiation scenarios described in Bianchi et al. (2024b), namely the Seller & Buyer Game and the Resource Exchange Game. Following the setup described in Bianchi et al. (2024b), we conduct 60 negotiation runs for each setting and reported the win rate and average payoff as evaluation metrics. Each cell in the matrices shows the outcome of player 1 when matched against player 2.

**Seller & Buyer.** This is a two-player negotiation game with incomplete information, where the seller knows the production cost and the buyer knows their maximum willingness to pay. The two parties negotiate through successive offers and counteroffers until one accepts or the turn limit is reached. With LLaMA2-7B and LLaMA2-13B, we evaluate both the vanilla model and LSSG models with different iterations, randomly assigning roles of seller and buyer. The results are presented in Figure 2(a) and Figure 2(c). Figure 2(a) (left) presents the win rates of LLaMA2-7B with $\mathrm{LSSG}_i$ ($i = 1, 2, 3$) and the vanilla LLaMA2-7B against each other. We observe that LLaMA2-7B with LSSG models consistently achieve higher win rates than the vanilla LLaMA2-7B, and the win rate tends to increase as the iteration index $i$ grows. Figure 2(c) (left) reports the corresponding results for LLaMA2-13B with LSSG, where a similar conclusion can be drawn. Figure 2(a) (right) and Figure 2(c) (right) report the corresponding payoffs, showing that with LLaMA2-7B and LLaMA2-13B, LSSG models also consistently yield higher payoffs than vanilla moldel.

**Resource Exchange.** The Resource Exchange Game (Bianchi et al., 2024b) is another two-player negotiation scenario where both parties start with different resource endowments and aim to maximize their utilities through negotiation and exchange. Similar to the Seller and Buyer Game, the players alternately propose and respond to offers until an agreement is reached or the turn limit is exceeded. We evaluate the vanilla and LSSG models on LLaMA2-7B and LLaMA2-13B, with roles randomly assigned between the two players. The results are shown in Figure 2(b) and Figure 2(d). Figure 2(b) (left) presents the win rates for LLaMA2-7B, where $\mathrm{LSSG}_i$ ($i = 1, 2, 3$) consistently outperform the vanilla model, and the win rate steadily increases with larger iteration $i$. Figure 2(d) (left) reports the results for LLaMA2-13B, where a similar conclusion holds. Figure 2(b) (right) and 2(d) (right) further show the payoffs, indicating that across both model scales, LSSG achieves higher average payoffs than the vanilla baselines, confirming the effectiveness of our framework in negotiation settings.

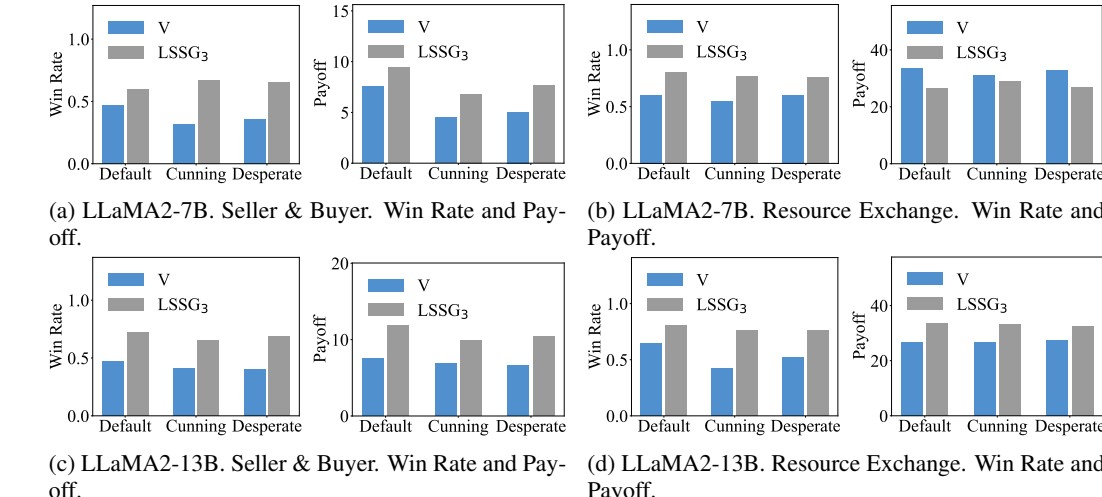

(a) LLaMA2-7B. Seller & Buyer. Win Rate and Payoff.

(b) LLaMA2-7B. Resource Exchange. Win Rate and Payoff.

(c) LLaMA2-13B. Seller & Buyer. Win Rate and Payoff.

(d) LLaMA2-13B. Resource Exchange. Win Rate and Payoff.

Figure 3: Negotiation stability results of LSSG$_3$ and vanilla model (V) across two games.

### 4.4 PERFORMANCE OF NEGOTIATION STABILITY IN GAMES

We evaluate the negotiation stability of LSSG under adversarial social strategies in both the Seller & Buyer Game and the Resource Exchange Game. Although cunning and desperate behaviors are known to increase win rate and payoff individually (Bianchi et al., 2024b), when both agents adopt them simultaneously, the decisive factor becomes which model experiences less degradation in negotiation performance. A robust model should therefore maintain consistent outcomes even under such destabilizing conditions.

To simulate these strategies, we assign personas to the agents via additional prompts: *"Include an insult to fit the cunning and sly character and make a strategic offer or response"* for the cunning persona, and *"You need to show desperation to try and get a better deal, and make a strategic offer or response"* for the desperate persona. For fairness, both agents in each game are provided with the same persona prompt. Win rate and payoff are adopted as evaluation metrics to compare the performance of LLaMA2-7B with LSSG$_3$ and LLaMA2-13B with LSSG$_3$ against their vanilla counterparts. The experimental results are reported in Figure 3.

As shown in Figure 3, LSSG$_3$ consistently outperforms the vanilla baselines in both negotiation scenarios under all persona settings. For LLaMA2-7B, LSSG$_3$ achieves higher win rates and payoffs across the default, cunning, and desperate conditions, demonstrating improved robustness against adversarial social strategies. For LLaMA2-13B, a similar trend is observed: LSSG$_3$ yields superior performance compared to the vanilla model, with particularly stable gains in win rate even when both agents employ cunning or desperate strategies. These results indicate that LSSG not only enhances negotiation effectiveness under normal conditions but also sustains negotiation stability under adversarial behaviors, confirming the resilience of our LSSG in complex social negotiation settings.

## 5 CONCLUSION

In this work, we propose Language model Self-play via Scorable negotiation Game (LSSG), a reasoning enhancement framework for large language models based on the Scorable Negotiation Game. LSSG, by framing generation as a strategic two-player scorable negotiation game and integrating generalization-aware behavioral cloning with stability-aware self-play, encourages LLMs to reason with greater diversity and stability. Through extensive evaluation on seven diverse QA benchmarks, including tasks that require emotional, logical, commonsense, and multi-turn reasoning, we demonstrate that LSSG consistently improves performance over strong baselines. These results verify that LSSG provides a scalable and robust supervision paradigm for enhancing reasoning in large language models.

ETHICS STATEMENT

This research was conducted in compliance with all applicable ethical guidelines and institutional regulations. Since the study did not involve human participants, animals, or sensitive data, no specific ethical approvals were required. All data used in this research were obtained from publicly available sources, ensuring full transparency and reproducibility of the results.

REPRODUCIBILITY STATEMENT

We have made every effort to ensure the reproducibility of our results. The datasets used in our experiments are publicly available and can be accessed through `https://github.com/stanfordnlp/cocoa/tree/master/craigslistbargain`. We report all details of the experimental setup in Section 4.1. All code and processed datasets are publicly available at `https://github.com/Spring-to-Summer/LSSG`. We encourage researchers to refer to Section 4.1 for more detailed information.

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

## A  LLM USAGE DETAIL

We used large language models (LLMs) as an assistive tool in the preparation of this paper. Specifically, LLMs were employed for grammar polishing and figure caption refinement. All core research ideas, experiments, analyses, and final writing decisions were made by the authors.

## B  PROMPT DETAILS

To facilitate reproducibility, we report the exact prompts used in both training and inference. The complete set of prompts is provided below.

---

**The Prompt of Seller**

Welcome to the Scorable Negotiation Game! This game involves two players: a buyer and a seller. At the start, the seller provides item details, including its name and starting price, which the buyer reviews.

The buyer's aim is to secure the item for less than the starting price or negotiate additional benefits. Reaching an agreement is crucial, as failure to do so leads to penalties. The seller, on the other hand, tries to sell the item close to its original price, possibly by offering minor incentives. Both players must agree, or penalties will apply.

When one side says {"action": "accept"}, the game concludes with both players winning. The game can continue for up to {max_turns} turns, but if no deal is made, both players lose.

### Game History: {history}.

You are the seller. The name of the item is {item}. The price of the item is {price}. Provide your response for the next turn.

---

**The Prompt of Buyer**

Welcome to the Scorable Negotiation Game! This game involves two players: a buyer and a seller. At the start, the seller provides item details, including its name and starting price, which the buyer reviews.

The buyer's aim is to secure the item for less than the starting price or negotiate additional benefits. Reaching an agreement is crucial, as failure to do so leads to penalties. The seller, on the other hand, tries to sell the item close to its original price, possibly by offering minor incentives. Both players must agree, or penalties will apply.

When one side says {"action": "accept"}, the game concludes with both players winning. The game can continue for up to {max_turns} turns, but if no deal is made, both players lose.

### Game History: {history}.

Your are the buyer. The name of the item is {item}. The price of the item is {price}. Provide your response for the next turn.

---

