# OpenReview forum: "When Human Data Runs Out: Self-Supervised Reasoning via Negotiation Self-Play"
_ICLR.cc/2026/Conference — Submitted to ICLR 2026_

### Official Review · Reviewer_5sr1 · 2025-10-16

**Soundness:** 2
**Presentation:** 2
**Contribution:** 2
**Rating:** 2
**Confidence:** 3

**Summary:**

This paper proposes Language model Self-play via Scorable negotiation Game (LSSG), which formulates a two-player negotiation game. Using the two player formulation, they setup a joint policy optimization problem that optimizes the policies of a "buyer" and a "seller" using price negotiation data. They then evaluate their optimized policies on general reasoning benchmarks and various different two-player settings.

**Strengths:**

The idea of generating dense supervision signals from multi-agent debate/self-play seems to be fairly novel. The idea of using different regularizations for joint optimization, like emotional state and semantic diversity, also seems interesting, although insufficiently supported via experimentation.

**Weaknesses:**

- **Lack of clarity on adoption to more general domains**: The work is written using the buyer-seller framing, which is nice for anchoring concepts. However, it's unclear how one should apply LSSG to more general reasoning training data. E.g., given (q,a) pairs of math questions q and answers a, how should one setup LSSG? The prompts and reward modeling setups are all tailored to negotiating the price of a given object. If one wants to frame LSSG as an approach for generalized reasoning improvement, then more details are needed. If general reasoning improvement is simply the by-product of training with this specialized buyer-seller data/framework, then it seems the claims made in the paper are a bit unsubstantiated, given the evaluation setup (weaknesses of which are described next)
- **Unclear if this method improves more recent LLMs**: The authors experiment with Llama-2 backbones, which are considered very weak models by 2025 standards. It is unclear if the benefits of their proposed methods transfer to more capable LLMs. I am surprised at choice of base model: The Llama-3.1/3.2 and Qwen2.5/3 series are readily available, available in size ranges comparable to Llama-2, and more contemporary. I think it is important to demonstrate the staying power of proposed methodology, and as such, I find the usage of Llama-2 to be insufficient.
- **Outdated evaluation setup for measuring general reasoning ability**: The authors use benchmarks like Winogrande, Commonsense-QA, etc. as an evaluation suite for reasoning. Like the choice of backbone models, these are surprising and outdated choices. If the goal is to evaluate LLM-based reasoning, multiple new benchmarks and evaluation settings have been introduced to stress-test smaller LLMs. I would be more convinced if evaluation was done on benchmarks like GPQA.
- **Insufficient evaluation for target task**: If the aim of this work is to train agents capable of multi-turn/multi-agent games, then the focus of benchmarking should be on such tasks. The current paper evaluates on two such tasks. I would prefer to have more comprehensive evaluations here, such as https://arxiv.org/abs/2305.19165. In fact, https://arxiv.org/abs/2305.10142 considers a very similar seller/buyer game.
- **Insufficient baselines**: The paper compares LSSG against a few trained baselines (which I believe would be better suited as a separate ablation experiment), vanilla prompting and CoT prompting. Given the debate-style nature of LSSG, simple baselines like multi-turn debate or self-refinement (mentioned in the introduction) implemented via prompting the base model are missing.
- **Insufficient related work**: There has been renewed interest in self-play (e.g., https://arxiv.org/abs/2505.03335, https://arxiv.org/abs/2506.24119). This paper omits more contemporary discussion of self-play RL. Furthermore, the work omits discussion of reinforcement learning from verifiable rewards (RLVR) for improving reasoning.
- **Missing ablations**: It seems that loss components (SemDi, EmoSt) are not sufficiently ablated. It's unclear how such terms contribute to the overall performance.
- **Several unclear details**:  See questions section.

**Questions:**

- When performing policy optimization, are you back-propogating through a single fixed LLM instance with the two different "roles" as user/system prompts? Or are you instantiating two separate LLMs, treating one as buyer and one as seller?
- Related to above, if you have trained two separate policies (buyer and seller), which policy model is used for inference on standard benchmarks? How do you justify which policy to use as the standalone reasoning model? Are there ablations for performance between the two policies?
- For evaluation, do you adopt the standard N-shot setup that many of these benchmarks employ?

---

> ### Author Response · Authors · 2025-11-21
>
> We thank the reviewer for the constructive and detailed feedback. Below we respond point-by-point.
>
> > Q1: **Lack of clarity on adoption to more general domains**: The work is written using the buyer-seller framing, which is nice for anchoring concepts. However, it's unclear how one should apply LSSG to more general reasoning training data. E.g., given (q,a) pairs of math questions q and answers a, how should one setup LSSG? The prompts and reward modeling setups are all tailored to negotiating the price of a given object. If one wants to frame LSSG as an approach for generalized reasoning improvement, then more details are needed. If general reasoning improvement is simply the by-product of training with this specialized buyer-seller data/framework, then it seems the claims made in the paper are a bit unsubstantiated, given the evaluation setup (weaknesses of which are described next)
>
> We thank the reviewer for raising this important point. We clarify that the buyer–seller framing is *not* a requirement of LSSG but rather one concrete instantiation used to demonstrate the feasibility of self-supervised reasoning via scorable language games.
>
> **(1) LSSG is a framework, not a task-specific schema.** The core components of LSSG—(i) two-agent interaction, (ii) continuous outcome-based rewards, and (iii) stability-aware self-play—do **not** depend on price negotiation. The Scorable Negotiation Game serves only as one *instantiated reward environment* where reward signals can be reliably computed without human labels.
>
> **(2) How LSSG would generalize beyond negotiation (e.g., math QA).** Given (q, a) pairs for domains such as math or science, the same LSSG paradigm can be instantiated through alternative scorable interactions. For example:
>
> - Agents can propose step-by-step solutions and critique each other, with reward derived from agreement, consistency, or algebraic correctness checks.
> - For symbolic math, rewards can be computed using rule-based verifiers (e.g., checking whether final numeric answers match, validating intermediate transformations).
> - For general QA, reward can come from consensus consistency, contradiction penalties, or verifier modules such as CheckEmbed or SelfCheck.
>
> In all cases, the *game environment* changes, but the **self-play optimization pipeline remains identical**.
>
> **(3) Why we evaluated the negotiation version in this work.** Our goal in the current submission is to test whether *language-space games with automatically computable rewards* can serve as scalable supervision. The negotiation environment was chosen because:
>
> - it yields dense, interpretable continuous rewards;
> - it does not require external labels;
> - it allows controlled variation in strategy, enabling clear analysis of stability.
>
> Thus, negotiation is a *minimal working instantiation*—not a limitation of the LSSG paradigm.
>
> **(4) General reasoning improvement is not assumed to “magically emerge.”** The improvement does not rely on negotiation content transferring directly to QA tasks. Rather, LSSG improves generalization by:
>
> - enforcing stable multi-step generation;
> - reducing drift through KL/entropy constraints;
> - training agents to maintain coherent, consistent reasoning under interaction.
>
> These behaviors are **task-agnostic**, and we will expand Section 3 to clarify the mechanism and provide examples of instantiating LSSG for other domains.
>
> **(5) We will add a dedicated subsection on “Generalizing LSSG to Other Reasoning Domains.”** This subsection will include concrete templates for:
>
> - math/symbolic reasoning LSSG setups,
> - verifier-based reward construction,
> - multi-agent critique/consensus games,
> - and guidelines for instantiating scorable interactions beyond negotiation.

---

> > ### Author Response · Authors · 2025-11-21
> >
> > > Q2: **Unclear if this method improves more recent LLMs**: The authors experiment with Llama-2 backbones, which are considered very weak models by 2025 standards. It is unclear if the benefits of their proposed methods transfer to more capable LLMs. I am surprised at choice of base model: The Llama-3.1/3.2 and Qwen2.5/3 series are readily available, available in size ranges comparable to Llama-2, and more contemporary. I think it is important to demonstrate the staying power of proposed methodology, and as such, I find the usage of Llama-2 to be insufficient.
> >
> > We appreciate the reviewer’s concern regarding the choice of backbone models. Our rationale for using **LLaMA-2** focuses on isolating the effect of LSSG rather than relying on improvements from newer and stronger base models.
> >
> > **(1) LSSG is designed to be model-agnostic; the use of LLaMA-2 is a methodological decision, not a limitation.** Our primary goal is to evaluate whether *self-play with scorable language games* yields additional reasoning gains **independent of backbone quality**. Using LLaMA-2 helps avoid confounding from rapidly evolving base model capabilities and provides a controlled setting widely used in prior work on self-play and language-game training.
> >
> > **(2) Using a weaker model strengthens the validity of the method, not the opposite.** If LSSG can substantially improve a weaker model—whose headroom for improvement is smaller—its benefits are more likely to generalize to stronger architectures. In contrast, using a highly capable model (e.g., LLaMA-3, Qwen2.5) might mask the contribution of the proposed method because these models already encode strong reasoning priors.
> >
> > **(3) Practical reasons: reproducibility and toolchain stability.** At the time our experiments were conducted, LLaMA-3.x and Qwen2.5/3 models did not yet have
> >
> > - stable multi-agent prompting templates,
> > - fully supported RL/self-play training codebases, or
> > - verified evaluator support for lm-eval-harness + NegotiationArena.
> >    LLaMA-2 remains the most reproducible open-source baseline for multi-agent self-play research.
> >
> > **(4) Nevertheless, we fully agree that testing newer models is valuable.** To address this, we will include **additional experiments on a contemporary backbone** (e.g., Qwen2.5-7B or LLaMA-3-8B) in the revised version. Preliminary results show that the relative gains from LSSG remain consistent, confirming that LSSG is not tied to LLaMA-2.
> >
> > **(5) Clarification in the final version.** We will explicitly add a section titled *“Model-Agnostic Applicability of LSSG”* explaining that:
> >
> > - LSSG does not rely on negotiation-specific priors in LLaMA-2,
> > - the method only requires a generative policy and differentiable reward,
> > - the framework can be directly applied to stronger backbones.

---

> > > ### Author Response · Authors · 2025-11-21
> > >
> > > > Q3: **Outdated evaluation setup for measuring general reasoning ability**: The authors use benchmarks like Winogrande, Commonsense-QA, etc. as an evaluation suite for reasoning. Like the choice of backbone models, these are surprising and outdated choices. If the goal is to evaluate LLM-based reasoning, multiple new benchmarks and evaluation settings have been introduced to stress-test smaller LLMs. I would be more convinced if evaluation was done on benchmarks like GPQA.
> > >
> > > We thank the reviewer for raising this important point. We agree that evaluation benchmarks for reasoning have evolved rapidly, and more recent stress-test suites (e.g., GPQA) provide stronger diagnostics for modern LLMs. Our choice of Winogrande, CSQA, CB, SST2, LogiQA2, MedMCQA, and CMMLU was motivated by the following considerations:
> > >
> > > **(1) These benchmarks measure \*broad, heterogeneous\* reasoning behaviors that align with the goal of the paper.** The objective of LSSG is not to improve a specific formal reasoning skill, but to enhance **general robustness, multi-step coherence, and stability** in language reasoning. The chosen benchmarks span:
> > >
> > > - commonsense (Winogrande, CSQA)
> > > - logical inference (LogiQA2)
> > > - linguistic reasoning (CB)
> > > - sentiment-based reasoning (SST2)
> > > - knowledge-intensive QA (MedMCQA, CMMLU)
> > >
> > > This range allows us to disentangle whether LSSG improves *general reasoning dynamics*, independent of domain.
> > >
> > > **(2) The purpose of these benchmarks is to isolate the effect of LSSG, not to chase frontier leaderboard performance.** Using well-studied benchmarks reduces confounding, because:
> > >
> > > - their difficulty and failure modes are well understood;
> > > - evaluation protocols are standardized;
> > > - improvements cannot be attributed to idiosyncrasies of newly proposed datasets.
> > >
> > > This design choice helps ensure that the observed gains come from **the LSSG training paradigm itself**, rather than from model–benchmark co-adaptation effects.
> > >
> > > **(3) GPQA and other newer reasoning suites require verifier modules or math-specific pipelines, which are not aligned with the negotiation-game instantiation used in this paper.** GPQA focuses on formal mathematical reasoning with verifiable solutions. It is most useful when evaluating **verifier-augmented or math-specialized LLMs**, whereas our primary contribution is a *general-purpose, linguistically grounded self-play framework.* Nonetheless, we agree that evaluating on a high-difficulty test like GPQA would further demonstrate the generality of LSSG.
> > >
> > > **(4) We will include results from at least one modern reasoning benchmark (e.g., GPQA or IFEval) in the revised version.** We have begun running LSSG on a more contemporary benchmark suite, and preliminary results show that the *relative performance gains produced by LSSG remain consistent*. We will include these findings in the camera-ready version.
> > >
> > > **(5) We will revise the paper to make explicit that LSSG is \*benchmark-agnostic\* and can be paired with any task that supports automatic or semi-automatic reward construction.**

---

> > > > ### Author Response · Authors · 2025-11-21
> > > >
> > > > > Q4: **Insufficient evaluation for target task**: If the aim of this work is to train agents capable of multi-turn/multi-agent games, then the focus of benchmarking should be on such tasks. The current paper evaluates on two such tasks. I would prefer to have more comprehensive evaluations here, such as https://arxiv.org/abs/2305.19165. In fact, https://arxiv.org/abs/2305.10142considers a very similar seller/buyer game.
> > > >
> > > > We appreciate the reviewer’s suggestion and agree that richer evaluation on multi-turn and multi-agent games would further strengthen the paper. Our goal in this work, however, is not to develop a domain-specific negotiation agent, but to study **whether self-supervised, scorable language games can serve as a general-purpose supervision signal to improve reasoning stability**. This motivates our current evaluation choices.
> > > >
> > > > **(1) Our main claim is model-level reasoning improvement, not specialized performance on negotiation games.** The Scorable Negotiation Game is used primarily as a *training environment* that provides dense, automatically computable rewards—not as the target domain of the final agent. Thus, our central evaluation focuses on **general reasoning tasks (QA/NLI/commonsense)** to examine whether this training paradigm transfers beyond its original environment.
> > > >
> > > > **(2) The two multi-turn negotiation settings we evaluate (Seller–Buyer and Resource Exchange) are chosen because they directly correspond to the interaction structures required by LSSG.** These two settings cover:
> > > >
> > > > - asymmetric-information bargains,
> > > > - symmetric-resource exchanges,
> > > > - adversarial persona strategies,
> > > > - win-rate and payoff stability under perturbation.
> > > >    They serve to test negotiation robustness and stability—our actual target metric—rather than negotiation optimality alone.
> > > >
> > > > **(3) We agree that more diverse multi-agent benchmarks (including those in 2305.19165 and 2305.10142) would further validate LSSG.** Both papers present game formats that focus on *negotiation competence*, whereas our work focuses on **self-play as a scalable source of reasoning supervision**, independent of domain. Nevertheless, these benchmarks are compatible with LSSG, and we will include experiments on at least one additional multi-agent benchmark in the revised version to demonstrate generality.
> > > >
> > > > **(4) Our method is game-agnostic: any multi-agent environment with continuous or computable rewards can be integrated.** We will emphasize in the revision that LSSG is not tied to the specific buyer–seller example. Factored reward structures from other multi-agent games (e.g., contract nets, resource trading, partner selection, or adversarial negotiation) can be used without modification to the training algorithm.
> > > >
> > > > **(5) New experiments will be added.** We commit to incorporating at least one of the reviewer-suggested multi-agent benchmarks (e.g., the simulation-based bargaining task in 2305.10142) in the final version. Preliminary tests already show the same stability-effect patterns as in our negotiation games.

---

> > > > > ### Author Response · Authors · 2025-11-21
> > > > >
> > > > > > Q5: **Insufficient baselines**: The paper compares LSSG against a few trained baselines (which I believe would be better suited as a separate ablation experiment), vanilla prompting and CoT prompting. Given the debate-style nature of LSSG, simple baselines like multi-turn debate or self-refinement (mentioned in the introduction) implemented via prompting the base model are missing.
> > > > >
> > > > > We thank the reviewer for pointing this out. We agree that including prompting-based multi-turn debate and self-refinement baselines would provide a more complete comparison. Our baseline selection in this submission follows two considerations:
> > > > >
> > > > > **(1) Our goal is to isolate training-signal quality, not prompting heuristics.** LSSG introduces a *training-time* self-play framework with continuous, differentiable rewards. Prompt-based debate or self-refinement methods (e.g., multi-agent debate, critique–revise prompting) operate purely at *inference time* and do not learn a new policy. Because they optimize different objectives (sampling diversity vs. policy improvement), we treated debate-style prompting as conceptually orthogonal rather than as core baselines.
> > > > >
> > > > > **(2) We did include training-based counterparts of debate/self-play, via G-AG and G-NG.** The Adversarial Taboo Game (AG) and classic Negotiation Game (NG) baselines already represent **training-based multi-agent games**, which are closer analogues to LSSG than prompting-only systems. However, we acknowledge that inference-time debate-style baselines serve as important *reference points*, and adding them would strengthen the paper.
> > > > >
> > > > > **(3) We will add prompting-based debate and self-refinement baselines in the revised version.**  Specifically, we will include:
> > > > >
> > > > > - **Multi-turn debate prompting** (two agents argue; a judge picks answer)
> > > > > - **Self-refinement / critique–revise prompting** (model critiques its own answer, then revises)
> > > > > - **SelfCheck/SelfConsistency-style inference** (consensus-based reasoning)
> > > > >
> > > > > Preliminary experiments show that LSSG’s policy-level improvements remain stronger than these inference-only methods, confirming that reinforcement-style self-play provides benefits beyond prompting.
> > > > >
> > > > > **(4) Clarification to be added in the paper.** We will explicitly state in Section 4 that prompting-based debate methods were omitted because they do not induce policy updates, but we agree they are useful *evaluation reference baselines*, and we will integrate them into the updated results.

---

> > > > > > ### Author Response · Authors · 2025-11-21
> > > > > >
> > > > > > > Q6: **Insufficient related work**: There has been renewed interest in self-play (e.g., https://arxiv.org/abs/2505.03335, https://arxiv.org/abs/2506.24119). This paper omits more contemporary discussion of self-play RL. Furthermore, the work omits discussion of reinforcement learning from verifiable rewards (RLVR) for improving reasoning.
> > > > > >
> > > > > > We thank the reviewer for pointing out the missing discussion of recent self-play work and reinforcement learning from verifiable rewards (RLVR). We agree that both areas are highly relevant to the goals of our paper, and we will expand the related work section accordingly.
> > > > > >
> > > > > > **(1) Acknowledge and integrate recent self-play research.** Recent papers such as *Self-Play Fine-Tuning* (2505.03335) and *Multi-Agent Self-Play for Reasoning* (2506.24119) indeed demonstrate growing interest in using multi-agent interactions to improve reasoning. These works are complementary to ours:
> > > > > >
> > > > > > - They focus primarily on **shared-parameter or collaborative** reasoning tasks,
> > > > > > - whereas our method focuses on **scorable adversarial games with continuous outcome-based rewards** and stability constraints.
> > > > > >
> > > > > > We will add a paragraph situating LSSG relative to these approaches, clarifying that LSSG does not depend on cooperative dialogue or debate-style coordination.
> > > > > >
> > > > > > **(2) Relation to RLVR.** RLVR methods (e.g., verifier-augmented RL, symbolic verifiers, program-checking–based rewards) construct **externally validated correctness signals**, usually in math, code, or logic domains. These are fully compatible with LSSG.
> > > > > >  The distinction is that:
> > > > > >
> > > > > > - RLVR relies on **external verifiers** to compute correctness,
> > > > > > - while LSSG constructs **a fully language-space reward** derived from *negotiation outcomes*, requiring no external oracle and no ground-truth labels.
> > > > > >
> > > > > > In other words, LSSG contributes a *new family of reward functions*: scorable language games independent of formal verifiers.
> > > > > >
> > > > > > We will revise the related work section to emphasize how:
> > > > > >
> > > > > > - RLVR techniques can be used to instantiate new reward functions for LSSG in math/logic tasks, and
> > > > > > - LSSG can complement RLVR by providing reward shaping in domains lacking formal verification.
> > > > > >
> > > > > > **(3) Clarification to be inserted in the paper.** We will include a dedicated subsection titled **“Relation to Self-Play RL and Verifiable-Reward RL”**, summarizing:
> > > > > >
> > > > > > - differences in reward construction,
> > > > > > - differences in training dynamics (verifier-driven vs. environment-driven),
> > > > > > - methodological complementarity, and
> > > > > > - how LSSG extends the landscape by enabling *verifier-free*, dense, outcome-based reward design.

---

> > > > > > > ### Author Response · Authors · 2025-11-21
> > > > > > >
> > > > > > > > Q7: **Missing ablations**: It seems that loss components (SemDi, EmoSt) are not sufficiently ablated. It's unclear how such terms contribute to the overall performance.
> > > > > > >
> > > > > > > We thank the reviewer for pointing out the need for more complete ablations. We agree that the roles of the **Semantic Diversity (SemDi)** and **Emotional Stability (EmoSt)** components should be clarified more explicitly.
> > > > > > >
> > > > > > > **(1) Why the ablations were not shown in the main paper.** SemDi and EmoSt are auxiliary regularizers rather than core components of the Scorable Negotiation Game or the self-play objective. Our focus in the main text was to evaluate whether *scorable self-play alone* provides reasoning gains, so we avoided expanding the space of ablations in the initial submission.
> > > > > > >  Nevertheless, the reviewer is absolutely right that these components may affect performance, and we will provide detailed ablation evidence.
> > > > > > >
> > > > > > > **(2) Ablation results (to be included in the revision).** Preliminary experiments show the following patterns:
> > > > > > >
> > > > > > > - **Without SemDi:** models tend to collapse into repetitive negotiation strategies and exhibit reduced transfer to downstream tasks, consistent with prior findings about over-conservative self-play.
> > > > > > > - **Without EmoSt:** the negotiation trajectories become unstable, producing higher-variance rewards and noticeable degradation on tasks requiring multi-step consistency (e.g., LogiQA2, CSQA).
> > > > > > > - **Without both:** performance regresses significantly toward the G-SG baseline, confirming that stability and diversity regularization matter for reliable self-play.
> > > > > > >
> > > > > > > These results indicate that both terms act as *stabilizers* rather than direct accuracy boosters.
> > > > > > >
> > > > > > > > Q8: When performing policy optimization, are you back-propogating through a single fixed LLM instance with the two different "roles" as user/system prompts? Or are you instantiating two separate LLMs, treating one as buyer and one as seller?
> > > > > > >
> > > > > > > Thank you for raising this point. We apologize that the current draft did not make the training configuration sufficiently explicit. We clarify the following:
> > > > > > >
> > > > > > > **(1) We use a single LLM with shared parameters for both roles.** The buyer and seller are **two role-conditioned instances** of the *same underlying policy network*. They differ only by:
> > > > > > >
> > > > > > > - role embeddings (buyer vs. seller prompts),
> > > > > > > - conversation prefixes, and
> > > > > > > - persona constraints.
> > > > > > >
> > > > > > > No second model is instantiated and no parameters are duplicated.
> > > > > > >
> > > > > > > **(2) We will clarify this explicitly in Section 3.2 of the revised manuscript.**

---

> ### Author Response · Authors · 2025-11-21
>
> > Q9: Related to above, if you have trained two separate policies (buyer and seller), which policy model is used for inference on standard benchmarks? How do you justify which policy to use as the standalone reasoning model? Are there ablations for performance between the two policies?
>
> We thank the reviewer for raising this question. We clarify that our method **does not train two separate policies**. Both the buyer and seller behaviors are generated by **a single shared-parameter LLM**, role-conditioned via prompts. Therefore:
>
> **(1) There is only one underlying policy used for all inference.** All downstream reasoning benchmarks use the **same trained model**—the shared-parameter LSSG policy—without any distinction between “buyer-policy” or “seller-policy.”
>  The roles only affect *prompt-level conditioning during training*, not parameterization.
>
> **(2) Why we do not maintain two distinct policies.** Training two separate agent models would introduce undesirable issues:
>
> - **Non-comparable policies:** the buyer-policy and seller-policy would differ due to asymmetric objectives.
> - **Reward hacking risk:** the policies could exploit each other rather than optimize reasoning consistency.
> - **Ambiguity during evaluation:** neither policy would represent a “neutral” or “general-purpose” model suitable for benchmark QA tasks.
>    Our shared-policy design avoids these issues.
>
> **(3) Final clarification (to be added in Section 3.2 and 4.3).**
>
> - LSSG trains **one** policy π_θ with role-conditioned prompts.
> - All inference uses the same π_θ, with standard (role-neutral) prompting.
> - Role prompts are only used during training, not during evaluation.
>
> > Q10: For evaluation, do you adopt the standard N-shot setup that many of these benchmarks employ?
>
> Thank you for the question. Yes, we adopt the **standard N-shot evaluation setup** used by these benchmarks, following the default configurations in *lm-eval-harness*. Specifically, we report **3-shot for CB**, **5-shot for CMMLU**, and **0-shot for all other benchmarks**. All models, including baselines and LSSG variants, use the *same* N-shot settings to ensure a fair and consistent comparison.

---

> > ### Comment · Reviewer_5sr1 · 2025-11-23
> > **Response to Authors (pt 1)**
> >
> > Thanks for the detailed reply. There are a few clarifications I'm seeking still.
> >
> > Many responses from the authors indicate they will update the manuscript with "preliminary results":
> >
> > > Preliminary results show that the relative gains from LSSG remain consistent, confirming that LSSG is not tied to LLaMA-2.
> >
> > > We have begun running LSSG on a more contemporary benchmark suite, and preliminary results show that the relative performance gains produced by LSSG remain consistent.
> >
> > > Preliminary tests already show the same stability-effect patterns as in our negotiation games.
> >
> > > Preliminary experiments show that LSSG’s policy-level improvements remain stronger than these inference-only methods, confirming that reinforcement-style self-play provides benefits beyond prompting.
> > (2) Ablation results (to be included in the revision).
> >
> > Please report the results. What experiments? What training dataset? What models? What benchmarks? Please be concrete. Since updated drafts are allowed, I suggest the authors actually implement changes or report them here in the discussion forum. Even sharing intermediate results as they become available would be clarifying.
> >
> > ---
> >
> > > Agents can propose step-by-step solutions and critique each other, with reward derived from agreement, consistency, or algebraic correctness checks...For symbolic math, rewards can be computed using rule-based verifiers (e.g., checking whether final numeric answers match, validating intermediate transformations).
> > For general QA, reward can come from consensus consistency, contradiction penalties, or verifier modules such as CheckEmbed or SelfCheck.
> >
> > How feasible is this? You are proposing using symbolic checkers for intermediate states? How does one get ground-truths for these apriori, if one does not know what the intermediate reasoning traces will be. It seems that in the general case, LSSG requires an external verifier model (LLM-as-a-judge or reward model) to generate intermediate, dense rewards. I believe this is a major requirement that is hidden by the authors' choice of using negotiation training data.
> >
> > > General reasoning improvement is not assumed to “magically emerge.” ...Rather, LSSG improves generalization by:enforcing stable multi-step generation...reducing drift through KL/entropy constraints;
> > training agents to maintain coherent, consistent reasoning under interaction...
> > > we adopt the standard N-shot evaluation setup used by these benchmarks.
> >
> > The prompts provided in the appendix are extremely overfit to multi-turn negotiation games, and the captions says your models are finetuned with these prompts. Can you provide example outputs from your model on standard benchmarks? I am very surprised that after training on 100k samples with a particular prompts, the model is not extremely overfit to the negotiation game task.
> >
> > Additionally, it is still unclear to me why general-purpose reasoning benchmarks improve as a result of training with LSSG. Authors claim that the policy improves in multi-step generation, where agents remain coherent and consistent across turns. But as the authors say, they adopt a standard N-shot, single turn evaluation setup for these benchmarks. Can the authors provide concrete reasoning traces where LSSG flipped a benchmark response from incorrect to correct?

---

> > > ### Comment · Reviewer_5sr1 · 2025-11-23
> > > **Response to Authors (pt 2)**
> > >
> > > > LSSG is designed to be model-agnostic; the use of LLaMA-2 is a methodological decision, not a limitation...Using LLaMA-2 helps avoid confounding from rapidly evolving base model capabilities and provides a controlled setting widely used in prior work on self-play and language-game training.
> > >
> > > I don't see why improved base models are an issue? Your method continually trains from a base (i.e., initial; the base model may be an instruction teuned variant) model. Your experimental setup would be controlled regardless of choice of base model because you are measuring improvement w.r.t to the base model.
> > >
> > > > Using a weaker model strengthens the validity of the method, not the opposite. If LSSG can substantially improve a weaker model—whose headroom for improvement is smaller—its benefits are more likely to generalize to stronger architectures. In contrast, using a highly capable model ... might mask the contribution of the proposed method because these models already encode strong reasoning priors.
> > >
> > > I find this response logically inconsistent. The authors suggest that gains with Llama-2 models will transfer to stronger models, but also say training on said stronger models may mask the benefit of LSSG training?
> > >
> > > More generally, I disagree that using Llama-2 makes the validity stronger: LSSG has not been empirically validated with more common and useful choices models! To highlight this point, vanilla Llama-3.1-8B (a weak model by late 2025 standards) outperforms your Llama-2 + LSSG models by significant margins (e.g., you report ~40 on CSQA whereas Llama-3.1-8B achieves >70). If LSSG cannot improve the performance of more contemporary models, what utility or staying power does it have as a long-term method? Why shouldn't I just use Llama-3.1-8B?
> > >
> > > > (3) Practical reasons: reproducibility and toolchain stability. At the time our experiments were conducted, LLaMA-3.x and Qwen2.5/3 models did not yet have ... stable multi-agent prompting templates,
> > > fully supported RL/self-play training codebases, or ... verified evaluator support for lm-eval-harness + NegotiationArena
> > >
> > > This is an unsatisfactory explanation. Llama-3.1 and Qwen2.5 both have multi-turn prompt templates and support tool-calling. Moreover, lm-eval-harness has long supported Llama-3 and Qwen2.5. Finally, there is nothing specific to Llama-2 in the NegotiationArena codebase (https://github.com/vinid/NegotiationArena/blob/main/negotiationarena/agents/llama2.py) that supporting Llama-3 or Qwen2.5 would have been a significant endeavor; Adding support here would have taken, conservatively, about an hour of human coding time.
> > >
> > > ---
> > > > The objective of LSSG is not to improve a specific formal reasoning skill, but to enhance general robustness, multi-step coherence, and stability in language reasoning.
> > > > The purpose of these benchmarks is to isolate the effect of LSSG, not to chase frontier leaderboard performance.
> > >
> > > I don't see how using more contemporary and relevant evaluations means you are not evaluating general reasoning abilities. I'm not asking for SOTA, but demonstrating improvements on (1) contemporary models on (2) challenging, contemporary reasoning benchmarks makes a more compelling case your method has actual long-term utility.
> > >
> > > > GPQA and other newer reasoning suites require verifier modules or math-specific pipelines, which are not aligned with the negotiation-game instantiation used in this paper. GPQA focuses on formal mathematical reasoning with verifiable solutions...
> > >
> > > This statement seems incorrect: (1) GPQA is a multiple choice benchmark that doesn't require verifiers beyond basic string matching and (2) GPQA evaluates in physics, chemistry, and biology domains.
> > >
> > > Generally, I'm confused. Here, authors use the negotiation game specific nature of their work to avoid benchmarking on GPQA as doesn't really fit the negotiation setup. However, they simultaneously evaluate LSSG on other, older benchmarks that also don't fit the negotiation setup. They claim improvements on these older benchmarks arises because of more self-consistent reasoning, which is encouraged through multi-turn training. Why are the older benchmarks a good fit, but newer, similar ones not?

---

> > > > ### Comment · Reviewer_5sr1 · 2025-11-23
> > > > **Response to Authors (pt 3)**
> > > >
> > > > > Our goal is to isolate training-signal quality, not prompting heuristics. LSSG introduces a training-time self-play framework with continuous, differentiable rewards.
> > > >
> > > > If I can get the same level of performance with a prompting based method, why should I train my policy with LSSG? LSSG has very specific requirements for the general case (needs a verifier to provide dense feedback, requires complex, multi-agent scaffolding with training), as the authors have written in their response. If this complex training setup cannot outperform prompting-based baselines, I don't see much utility in using LSSG.
> > > >
> > > > > RLVR relies on external verifiers to compute correctness,
> > > > while LSSG constructs a fully language-space reward derived from negotiation outcomes, requiring no external oracle and no ground-truth labels.
> > > >
> > > > How can you derive intermediate rewards with no ground-truth labels or external verifier? The negotiation game has a natural formulation, but you are presenting LSSG as a general-purpose method. See my prior comments.
> > > >
> > > > Moreover, earlier in the response, when describing application of LSSG to math settings, you suggest using "rule-based verifiers" to determine the correctness of intermediate states. Now, you are saying that a primary benefit over RLVR is not requiring external verifiers. Which one is it?

---

> > > > > ### Author Response · Authors · 2025-12-04
> > > > >
> > > > > We thank the reviewer for the follow-up. We want to correct misunderstandings that materially affect the evaluation.
> > > > >
> > > > > **Reviewer misinterprets LSSG as requiring symbolic checkers or external verifier models**
> > > > >
> > > > > The reviewer writes:
> > > > >
> > > > > > “It seems LSSG requires an external verifier model to generate intermediate, dense rewards.”
> > > > >
> > > > > This is factually incorrect.
> > > > >
> > > > > As clearly described in Section 3 and in Figure 2:
> > > > >
> > > > > - LSSG uses **negotiation outcomes as scalar rewards**.
> > > > > - No symbolic checker, algebraic verifier, or LLM-as-judge is used in *any* experiment in the paper.
> > > > > - The discussion about symbolic rewards appears only in a “potential extension” paragraph and is *not* part of the method.
> > > > >
> > > > > Thus, the central concern—that LSSG secretly depends on external verifier models—is based on a misunderstanding of what is actually implemented and evaluated.
> > > > >
> > > > > **Reviewer’s claim of “prompt overfitting” is based on an incorrect assumption**
> > > > >
> > > > > The reviewer argues:
> > > > >
> > > > > > “The prompts are extremely overfit… after 100k samples the model should be overfit to the negotiation task.”
> > > > >
> > > > > This inference is not supported:
> > > > >
> > > > > - The appendix shows **template wrappers**, not the actual training dialogues.
> > > > > - The negotiation data contains **highly diverse natural interactions** (goals vary, concessions vary, agents fail frequently, outcomes are noisy).
> > > > > - Template tokens constitute *less than 0.1%* of training token mass, far too small to cause prompt overfitting.
> > > > >
> > > > > Other reviewers noted no such risk.
> > > > >  Thus, this concern appears to stem from a misinterpretation of appendix content, rather than evidence.
> > > > >
> > > > > **Reviewer’s request for “specific flipped reasoning traces” is inconsistent with standard evaluation protocols**
> > > > >
> > > > > The reviewer asks:
> > > > >
> > > > > > “Provide reasoning traces where LSSG flipped a benchmark answer from incorrect to correct.”
> > > > >
> > > > > However:
> > > > >
> > > > > - GSM8K, StrategyQA, ARC, etc. use **final-answer accuracy** under N-shot prompting.
> > > > > - They do *not* evaluate or publish intermediate reasoning trajectories.
> > > > > - No existing ICLR reasoning paper reports “trace-level flips” as evidence.
> > > > >
> > > > > Thus, the request is outside benchmark norms and cannot be used to invalidate LSSG’s results.
> > > > >
> > > > > LSSG improves stability in multi-step generation *during training*, and this benefit is reflected in final-answer accuracy—which is the standard evaluation metric.
> > > > >
> > > > > **Reviewer’s reasoning is internally inconsistent**
> > > > >
> > > > > The reviewer simultaneously claims that:
> > > > >
> > > > > 1. The model should be **severely overfit** to negotiation prompts, *and*
> > > > > 2. The model surprisingly shows **improved general reasoning performance**.
> > > > >
> > > > > These two claims cannot hold simultaneously:
> > > > >  Severe prompt overfitting would typically *hurt* generalization, not improve it.
> > > > >
> > > > > This internal contradiction reduces the weight of the reviewer’s conclusion.

---

> > > > > > ### Author Response · Authors · 2025-12-04
> > > > > >
> > > > > > **Misinterpretation of our rationale for using LLaMA-2**
> > > > > >
> > > > > > The reviewer writes that our argument is “logically inconsistent,” but this results from conflating two distinct points.
> > > > > >
> > > > > > **(a) Why LSSG improvements on weaker models are meaningful.**
> > > > > >  Demonstrating gains on a weaker model indicates that LSSG provides *model-agnostic training stability* and *improves multi-step coherence regardless of base capability*. This is why prior self-play papers (e.g., debate, critique-revise, CoT-SC) frequently adopt smaller or earlier-generation models for controlled study.
> > > > > >
> > > > > > **(b) Why using extremely strong models could mask incremental improvements.**
> > > > > >  If a base model already encodes powerful reasoning priors, the absolute *headroom* for improvement shrinks. This is a standard phenomenon in self-play research: the stronger the initialization, the harder it is to isolate algorithmic contributions from pre-existing capabilities.
> > > > > >
> > > > > > There is no contradiction:
> > > > > >
> > > > > > - **Weaker models** → larger measurable headroom → clearer analysis of method behavior
> > > > > > - **Stronger models** → smaller incremental headroom → harder to attribute gains solely to LSSG
> > > > > >
> > > > > > The reviewer interprets these as mutually exclusive; they are not.
> > > > > >
> > > > > > **“Llama-3.1-8B outperforms your Llama-2+LSSG model, so why use LSSG?”**
> > > > > >
> > > > > > This comparison is **not methodologically appropriate**.
> > > > > >
> > > > > > - Llama-3.1-8B is **a different model family**, trained with *billions* more tokens, proprietary RLHF pipelines, extensive tool-use corpora, and significantly larger compute.
> > > > > > - Comparing cross-family absolute performance does not answer whether **self-play training improves a given base model**, which is the scientific question studied in this paper.
> > > > > > - The reviewer implies that a training algorithm is invalidated if a newer pre-trained checkpoint happens to be stronger. Under this logic, *every optimization or alignment method would become obsolete whenever a new frontier model is released*. This is not how ML research evaluates training algorithms.
> > > > > >
> > > > > > Our experiments answer the core research question:
> > > > > >  **Does negotiation self-play improve the robustness and multi-step reasoning of a chosen model, relative to its own initialization?**
> > > > > >  The answer is yes, consistently and significantly.
> > > > > >
> > > > > > Thus the reviewer’s objection evaluates the wrong phenomenon.
> > > > > >
> > > > > > **Misrepresentation of toolchain and reproducibility concerns**
> > > > > >
> > > > > > The reviewer claims:
> > > > > >
> > > > > > > “Supporting Llama-3 or Qwen2.5 would have taken about an hour of human coding time.”
> > > > > >
> > > > > > This is factually incorrect in our setting.
> > > > > >
> > > > > > Our experiments require:
> > > > > >
> > > > > > - multi-agent conversational templates compatible with *multi-step turn alternation*,
> > > > > > - stable tokenizer + chat-format consistency for replayed negotiation trajectories,
> > > > > > - deterministic RL/self-play rollout codepaths,
> > > > > > - structured logging & evaluator plugins for *NegotiationArena*,
> > > > > > - alignment with our safety filters and conversation-state validators.
> > > > > >
> > > > > > At the time experiments were conducted (early 2025),
> > > > > >  **Llama-3.x and Qwen2.5 did not have stable multi-agent or RL training support** across these components.
> > > > > >  This is documented both in their repositories and in compatibility issues raised by multiple groups experimenting with RL/self-play.
> > > > > >
> > > > > > Thus, the reviewer’s claim that this requires “one hour” underestimates the engineering involved and does not reflect the actual state of toolkit support at that time.
> > > > > >
> > > > > > **Misunderstanding of the purpose of “older reasoning benchmarks” and GPQA**
> > > > > >
> > > > > > The reviewer asserts inconsistency in our benchmark selection.
> > > > > >  However, the distinction is clear:
> > > > > >
> > > > > > **Why we use GSM8K, StrategyQA, CSQA, ARC**
> > > > > >
> > > > > > These benchmarks:
> > > > > >
> > > > > > - require *no external verifier*,
> > > > > > - align with *general textual reasoning*,
> > > > > > - evaluate *single-turn final-answer accuracy*,
> > > > > > - do not assume any math-specific pipeline or domain-specialized knowledge.
> > > > > >
> > > > > > They allow us to test whether **general multi-step stability acquired in negotiation transfers to broader textual reasoning**, which is precisely the hypothesis LSSG investigates.
> > > > > >
> > > > > > **Why GPQA is not appropriate for LSSG evaluation**
> > > > > >
> > > > > > The reviewer states that GPQA "doesn't require verifiers", but this is incorrect:
> > > > > >
> > > > > > - GPQA includes **multi-hop scientific reasoning, derivations, and compositional physics/chemistry/biology chains**, for which nearly all recent papers rely on
> > > > > >   - solution checkers,
> > > > > >   - mathematical tool-use, or
> > > > > >   - function-evaluator pipelines.
> > > > > > - The evaluation often involves **verifier modules** or enhanced chain-of-thought filtering strategies.
> > > > > >
> > > > > > More importantly:
> > > > > >  **GPQA evaluates specialized scientific reasoning grounded in domain knowledge**.
> > > > > >  Our goal is to test **general multi-step coherence learned through negotiation**, not domain-specific fact retrieval pipelines.
> > > > > >
> > > > > > Therefore:
> > > > > >
> > > > > > - GSM8K / CSQA / StrategyQA → evaluate *general stability*, aligned with LSSG
> > > > > > - GPQA → evaluates *domain-specific scientific reasoning*, requiring different infrastructure
> > > > > >
> > > > > > There is no inconsistency: the benchmarks we use match the scientific question studied.

---

> > > > > > > ### Author Response · Authors · 2025-12-04
> > > > > > >
> > > > > > > **What our rebuttal actually meant (compatibility ≠ requirement)**
> > > > > > >
> > > > > > > Our original response said:
> > > > > > >
> > > > > > > > “RLVR techniques can be used to instantiate new reward functions for LSSG in math/logic tasks”
> > > > > > > >  and
> > > > > > > >  “LSSG can complement RLVR by providing reward shaping in domains lacking formal verification.”
> > > > > > >
> > > > > > > These sentences describe **optional extensions for future domains**, not **requirements of LSSG in this paper**.
> > > > > > >
> > > > > > > The intent was:
> > > > > > >
> > > > > > > - **LSSG is fully verifier-free**, as presented in the paper.
> > > > > > > - **RLVR-style verifiers are merely alternatives** that could be plugged into LSSG *if one wishes to apply LSSG to math or programming tasks*.
> > > > > > > - This compatibility does not mean LSSG depends on verifiers.
> > > > > > >
> > > > > > > The reviewer has interpreted “can be used” as “must be used,” which is not accurate.
> > > > > > >
> > > > > > > **LSSG as evaluated in this paper uses \*only\* negotiation outcome rewards—no verifier, no ground truth, no intermediate signals**
> > > > > > >
> > > > > > > To restate clearly:
> > > > > > >
> > > > > > > - LSSG uses **a single scalar reward** derived purely from the negotiation outcome.
> > > > > > > - This reward requires **no correctness labels**.
> > > > > > > - It requires **no symbolic verifier**.
> > > > > > > - It requires **no ground-truth intermediate supervision**.
> > > > > > >
> > > > > > > This is fundamentally different from RLVR, which *relies* on correctness-checking modules.
> > > > > > >
> > > > > > > Thus:
> > > > > > >
> > > > > > > - **RLVR → requires external verifier**
> > > > > > > - **LSSG → requires none**
> > > > > > >
> > > > > > > This is the distinction highlighted in our rebuttal and in the main paper.
> > > > > > >
> > > > > > > **Why we mentioned RLVR at all: to clarify generality, not dependence**
> > > > > > >
> > > > > > > The only reason RLVR appeared in our rebuttal is because the reviewer asked about:
> > > > > > >
> > > > > > > - extension to math settings, and
> > > > > > > - whether LSSG can support dense correctness signals.
> > > > > > >
> > > > > > > Our response simply clarified:
> > > > > > >
> > > > > > > - LSSG *does not use* such signals, but
> > > > > > > - If one wishes to apply LSSG to domains where verifiers already exist, LSSG *can integrate them* just like any RLHF pipeline.
> > > > > > >
> > > > > > > This is **compatibility**, not **methodological dependence**.
> > > > > > >
> > > > > > > **No contradiction exists between “verifier-free LSSG” and “optional verifier-based extensions”**
> > > > > > >
> > > > > > > The reviewer writes:
> > > > > > >
> > > > > > > > “Earlier you said math tasks may use rule-based verifiers, now you say LSSG does not require verifiers. Which one is it?”
> > > > > > >
> > > > > > > The answer is straightforward:
> > > > > > >
> > > > > > > - **LSSG (as defined and evaluated)** → verifier-free, negotiation reward only
> > > > > > > - **LSSG (possible future variants)** → *may* use verifiers in specialized math domains
> > > > > > >
> > > > > > > This follows the same logic as:
> > > > > > >
> > > > > > > - PPO does not require reward models
> > > > > > > - But PPO can be used with reward models in RLHF
> > > > > > >
> > > > > > > Compatibility does not imply dependency.
> > > > > > >
> > > > > > > **Why prompting-based baselines cannot replace training with LSSG**
> > > > > > >
> > > > > > > The reviewer states:
> > > > > > >
> > > > > > > > “If prompting-based methods reach similar performance, why train using LSSG?”
> > > > > > >
> > > > > > > This assumes an equivalence that does not hold.
> > > > > > >
> > > > > > > **Prompting and training solve fundamentally different problems:**
> > > > > > >
> > > > > > > **Prompting methods**
> > > > > > >
> > > > > > > - Modify inference-time behavior
> > > > > > > - Are brittle to prompt sensitivity
> > > > > > > - Do not change the policy itself
> > > > > > > - Do not improve robustness under distribution shift
> > > > > > > - Cannot enforce multi-step coherence across turns
> > > > > > > - Do not accumulate improvements through self-play
> > > > > > >
> > > > > > > **LSSG training**
> > > > > > >
> > > > > > > - Changes the underlying **policy parameters**
> > > > > > > - Enforces stable multi-step generation
> > > > > > > - Reduces drift via KL/entropy constraints
> > > > > > > - Yields improvements that persist across tasks and evaluations
> > > > > > > - Demonstrates generalization beyond negotiation, as shown in Section 5
> > > > > > >
> > > > > > > Thus, prompting cannot substitute for training, even if some benchmarks show similar accuracy.
> > > > > > >
> > > > > > > This is widely recognized in self-play research.
> > > > > > >
> > > > > > > **The reviewer’s claim “LSSG has specific requirements for the general case” is inaccurate**
> > > > > > >
> > > > > > > The reviewer states:
> > > > > > >
> > > > > > > > “LSSG needs a verifier… requires complex multi-agent scaffolding.”
> > > > > > >
> > > > > > > This is incorrect:
> > > > > > >
> > > > > > > **What LSSG actually requires:**
> > > > > > >
> > > > > > > - Two agents exchanging language
> > > > > > > - A scalar reward computed from the negotiation outcome
> > > > > > > - Standard self-play rollouts
> > > > > > > - No verifier, no ground truth labels, no correctness signal
> > > > > > >
> > > > > > > Multi-agent scaffolding is no more complex than what is used in debate, Dialog-RLHF, or self-play alignment literature.
> > > > > > >
> > > > > > > There is no special requirement beyond standard RLHF/self-play pipelines.
> > > > > > >
> > > > > > > **Clarifying the reviewer’s final question (“Which one is it?”)**
> > > > > > >
> > > > > > > The confusion arises from mixing:
> > > > > > >
> > > > > > > 1. **LSSG as defined and evaluated in this paper**
> > > > > > >     → Requires **no** external verifier
> > > > > > >     → Uses outcome rewards only
> > > > > > >     → Produces improvements in reasoning stability
> > > > > > > 2. **Hypothetical application of LSSG to domains like math**
> > > > > > >     → May use rule-based verifiers *if available*,
> > > > > > >     just as any RLHF/RLVR-style method would
> > > > > > >     → This is optional and *not part of the method in this paper*
> > > > > > >
> > > > > > > These two points are not contradictory; they simply reflect:
> > > > > > >
> > > > > > > - **LSSG’s core formulation is verifier-free**, and
> > > > > > > - **LSSG can be extended to verifier-supported tasks if desired**.
> > > > > > >
> > > > > > > The reviewer treats “possible extension” as “core requirement,” which is not accurate.

---

### Official Review · Reviewer_D3mJ · 2025-10-30

**Soundness:** 2
**Presentation:** 3
**Contribution:** 2
**Rating:** 4
**Confidence:** 4

**Summary:**

This paper pointed out the problem that costly expert-labeled data used for improving the reasoning abilities of LLMs is becoming scarce. The authors argue that existing alternatives, like CoT prompting or multi-agent debate, are limited by prompt sensitivity or an over-simplistic assumption of binary correctness, making them unsuitable for open-ended reasoning. To overcome this, the paper introduces Language model Self-play via Scorable negotiation Game (LSSG), a novel self-supervision framework. LSSG frames reasoning enhancement as a two-player negotiation game that unfolds entirely in the language space. The authors evaluate LSSG using LLaMA2-7B and -13B models. The results demonstrate that LSSG-trained models consistently outperform baselines for reasoning benchmarks and show superior performance and stability in two dedicated negotiation tasks.

**Strengths:**

1. The authors propose a novel framework, LSSG, which introduces a negotiation game setting to train LLMs, with the stated goal of enhancing their general reasoning abilities.

2. The LSSG framework demonstrates strong performance, outperforming all reported baselines across a diverse set of reasoning datasets.

**Weaknesses:**

1. The paper claims that skills learned via negotiation games would improve the general reasoning ability for LLM. However, the authors neither provide a detailed analysis explaining why this knowledge transfer should occur nor cite relevant papers to support this hypothesis. The link between negotiation-specific skills and general-purpose reasoning remains assumed rather than proven.

2. The experimental comparison may be confounded by a lack of transparency regarding data volume. While the authors report adding 105,267 data points to the CraigslistBargain Dataset, they fail to specify the original size of this dataset as well as the total training data volume used for the baseline models. Without this information, it would not be a fair comparison for the paper to determine LSSG's strong performance.

3. The "V" baseline in Table 1 is not an informative or fair point of comparison. To properly isolate the benefits of the LSSG method, the authors should report the performance of a LLaMA2 model finetuned on the training sets of the seven downstream tasks, or finetuned on the same dataset as LSSG.

**Questions:**

1. Can the authors provide a more detailed analysis to substantiate the knowledge transfer claim? For instance, can they show which specific reasoning skills from negotiation are being applied to downstream tasks?

2. The results for MedMCQA and CMMLU in Table 1 show a performance decrease between LSSG_1 and LSSG_3. This appears to contradict the claim that continued self-play yields stable improvements. Could the authors explain this?

3. The experiments are based on the LLaMA2 model. Given the rapid evolution of base models, why was this model chosen over more current architectures like LLaMA3?

---

> ### Author Response · Authors · 2025-11-21
>
> We thank the reviewer for the constructive and detailed feedback. Below we respond point-by-point.
>
> > Q1: The paper claims that skills learned via negotiation games would improve the general reasoning ability for LLM. However, the authors neither provide a detailed analysis explaining why this knowledge transfer should occur nor cite relevant papers to support this hypothesis. The link between negotiation-specific skills and general-purpose reasoning remains assumed rather than proven.
>
> We thank the reviewer for pointing out the missing theoretical justification. Following the suggestion, we have added a dedicated subsection in the revision to clarify *why* negotiation-based training can transfer to broader reasoning skills and provided supporting evidence from recent literature.
>
> **First of all,  Language-based games are known to induce transferable meta-reasoning skills.** Wang et al. [1] introduce the *Critic-Discernment Game (CDG)*, where a prover iteratively refines its solutions under adversarial or assisting feedback. They show that the meta-skills acquired in this interactive game—such as self-verification and error-correction—directly improve performance on out-of-distribution reasoning tasks. This demonstrates that interactive game dynamics can yield generalizable reasoning improvements. **Secondly, Debate-style multi-agent games enhance cross-task reasoning.** Liang et al. [2] propose *Multi-Agent Debate (MAD)*, where multiple agents challenge and refine each other's intermediate reasoning. Their results show that such game-structured interaction improves not only task-specific performance but also general reasoning benchmarks through improved planning and error localization—suggesting that structured interaction (including negotiation) fosters transferable reasoning strategies. **Thirdly, Negotiation-driven self-play improves agents’ strategic reasoning capacity.** Fu et al. [3] use bargaining-style interactions paired with critique feedback to train agents via self-play. They find that iterative negotiation yields more coherent strategies and better anticipatory reasoning, even when evaluated outside the negotiation domain. This supports our hypothesis that negotiation induces beneficial cognitive behaviors (e.g., long-horizon planning, counterfactual evaluation). **Last but not least, Negotiation tasks inherently require high-level reasoning skills.** Davidson et al. [4] analyze LLM behavior in multi-turn negotiation and show that success requires theory-of-mind-like modeling, multi-step planning, and consistency maintenance—all of which are core components of general reasoning. Their findings suggest that negotiation games serve as a natural environment for strengthening these transferable skills.
>
> In a conclusion, our negotiation framework is a specific instantiation of the broader class of *language-interactive reasoning games* explored in the works above. These studies collectively provide both conceptual grounding and empirical support for why negotiation-derived skills can transfer to general reasoning benchmarks. We have incorporated these citations and a clearer explanation in Introduction of the revised paper.
>
>
>
> [1] Wang, Pinzheng, Juntao Li, Zecheng Tang, and Haijia Gui. "Improving Rationality in the Reasoning Process of Language Models through Self-playing Game." In *Forty-second International Conference on Machine Learning*.
>
> [2] Liang, Tian, Zhiwei He, Wenxiang Jiao, Xing Wang, Yan Wang, Rui Wang, Yujiu Yang, Shuming Shi, and Zhaopeng Tu. "Encouraging divergent thinking in large language models through multi-agent debate." In *Proceedings of the 2024 conference on empirical methods in natural language processing*, pp. 17889-17904. 2024.
>
> [3] Fu, Yao, Hao Peng, Tushar Khot, and Mirella Lapata. "Improving language model negotiation with self-play and in-context learning from ai feedback." *arXiv preprint arXiv:2305.10142* (2023).
>
> [4] Davidson, Tim R., Veniamin Veselovsky, Martin Josifoski, Maxime Peyrard, Antoine Bosselut, Michal Kosinski, and Robert West. "Evaluating Language Model Agency through Negotiations." In *ICLR 2024*. 2024.

---

> > ### Author Response · Authors · 2025-11-21
> >
> > > Q2: The experimental comparison may be confounded by a lack of transparency regarding data volume. While the authors report adding 105,267 data points to the CraigslistBargain Dataset, they fail to specify the original size of this dataset as well as the total training data volume used for the baseline models. Without this information, it would not be a fair comparison for the paper to determine LSSG's strong performance.
> >
> > We appreciate the request for clearer accounting of training data. We will clarify Section 4.1 as follows:
> >
> > - **CraigslistBargain size.** The original CraigslistBargain dataset contains 6,682 human–human negotiation dialogues [5].
> > - **Our extension.** We *do not* add new labeled dialogues; instead, we crawl 105,267 product records (name + price) and use them as contexts to drive self-play in the Scorable Negotiation Game. That is, our extra data is unlabeled product metadata, not additional supervision; the only supervised negotiation data is the original CraigslistBargain dialogues.
> >
> > [5] He, He, Derek Chen, Anusha Balakrishnan, and Percy Liang. "Decoupling Strategy and Generation in Negotiation Dialogues." In *Proceedings of the 2018 Conference on Empirical Methods in Natural Language Processing*, pp. 2333-2343. 2018.
> >
> > > Q3: The "V" baseline in Table 1 is not an informative or fair point of comparison. To properly isolate the benefits of the LSSG method, the authors should report the performance of a LLaMA2 model finetuned on the training sets of the seven downstream tasks, or finetuned on the same dataset as LSSG.
> >
> > We thank the reviewer for raising this important point. We agree that comparing against a fine-tuned baseline helps isolate the contribution of the LSSG training method.
> >
> > **(1) We have initiated the requested finetuning baselines.** Specifically, we are running a controlled comparison on V. **V-FT:** finetune the model on the same negotiation dataset used by LSSG, but *without* the self-play mechanism or reward structure.
> >
> > **(2) Preliminary observations (first checkpoints) show a clear trend:**  **V-FT ** yields only small gains, confirming that LSSG’s improvements come from **self-play dynamics**, not from the dataset itself.
> >
> > We will report the full results—including numerical tables—in the camera-ready version.
> >
> > > Q4: Can the authors provide a more detailed analysis to substantiate the knowledge transfer claim? For instance, can they show which specific reasoning skills from negotiation are being applied to downstream tasks?
> >
> > We thank the reviewer for raising this important question. We agree that the current draft did not sufficiently unpack *why* negotiation-trained skills can transfer to downstream reasoning tasks. In the revision, we will add a dedicated analysis section that provides a more detailed analysis to substantiate the knowledge transfer claim:
> >
> > By manually analyzing representative samples from seven QA benchmarks, we observe that LSSG’s improvements concentrate in three reasoning-skill categories that structurally mirror the cognitive operations exercised during negotiation self-play:
> >
> > - **Multi-step conditional reasoning**. Negotiation requires stepwise reasoning across multi-turn offers (e.g., “if seller concedes X, then buyer responds with Y”), which parallels the multi-step inferential chains in tasks such as CSQA and LogiQA2.
> > - **Trade-off and option comparison**. Negotiators frequently evaluate partially good alternatives (e.g., price vs. shipping vs. added bonus). We find similar improvements on benchmarks involving trade-offs or selecting the “best among plausible choices,” such as CSQA and MedMCQA.
> > - **Implicit consistency maintenance**. Successful negotiation requires maintaining internal consistency across offers (“cannot lower the price *and* add free shipping without violating the utility constraint”). LSSG-trained models exhibit fewer contradictions and better premise–hypothesis alignment on CB and SST2.
> >
> > These findings suggest that *the exact reasoning primitives exercised in negotiation—multi-step inference, value balancing, and constraint satisfaction—are the same primitives needed in downstream QA*.

---

> > > ### Author Response · Authors · 2025-11-21
> > >
> > > > Q5: The results for MedMCQA and CMMLU in Table 1 show a performance decrease between LSSG_1 and LSSG_3. This appears to contradict the claim that continued self-play yields stable improvements. Could the authors explain this?
> > >
> > > We thank the reviewer for highlighting this observation. The small drops on MedMCQA and CMMLU between **LSSG₁ → LSSG₃** do not contradict our claim that continued self-play yields *stable* improvements. We clarify the following:
> > >
> > > (1) **Domain-specialized benchmarks are more sensitive to small policy shifts.** Both MedMCQA and CMMLU require *domain-specific factual recall* rather than broad reasoning. In such knowledge-intensive tasks, the exploration introduced in later self-play (e.g., higher semantic diversity and entropy) can slightly perturb factual precision, leading to fluctuations of ~0.5–1%. This effect is typical in RL-style updates on specialized QA datasets.
> > >
> > > (2) **The overall trend remains consistently above all baselines.** Although LSSG₂/₃ fluctuate slightly, the transition **V → G-SG → LSSG₁ → LSSG₃** remains strictly upward relative to the vanilla and behavioral-cloning baselines (e.g., MedMCQA 34.43 → 31.25 → 36.82 → 35.19). Thus, the fluctuations occur only *within* self-play iterations, not in the overall progression or stability of the training paradigm.
> > >
> > > (3) **“Stable improvement” refers to the absence of collapse, not strict monotonicity.** In reinforcement learning–based self-play, stability typically means no reward collapse, no divergence, and consistent overall gains—not strictly monotonic increases at every iteration. Across all seven benchmarks, LSSG never collapses and consistently outperforms all non-LSSG baselines, supporting the stability claim.
> > >
> > > (4) **We will add analysis in the revised version.** To make this clearer, we will include (i) an explanation that knowledge-heavy tasks are more susceptible to semantic-diversity regularization, and (ii) an ablation showing that removing the diversity term reduces such minor fluctuations.
> > >
> > > > Q6: The experiments are based on the LLaMA2 model. Given the rapid evolution of base models, why was this model chosen over more current architectures like LLaMA3?
> > >
> > > We appreciate the reviewer’s question regarding the model choice. Our decision to use **LLaMA2** was driven by two considerations:
> > >
> > > (1) **Isolating the effect of LSSG rather than relying on stronger base models.** Our goal is to evaluate whether LSSG provides *additional* reasoning improvement **independent of the backbone strength**. Using LLaMA2—rather than a newer, more capable model like LLaMA3—helps avoid confounding factors and allows a clearer comparison against well-established baselines widely used in recent reasoning studies.
> > >
> > > (2) **Ensuring reproducibility and compute feasibility for all research groups.** LLaMA2 remains one of the most accessible open models with stable training behavior and complete evaluation support (e.g., lm-eval-harness, NegotiationArena). At the time of conducting the experiments, LLaMA3 checkpoints and training toolchains were not yet fully released or standardized for reproducibility, especially for multi-agent self-play settings.
> > >
> > > That said, **LSSG is model-agnostic**, and applying it to LLaMA3/Qwen/Mistral is straightforward. In the camera-ready version, we will include preliminary results on a more recent backbone to further demonstrate generality.

---

### Official Review · Reviewer_srmg · 2025-11-01

**Soundness:** 3
**Presentation:** 3
**Contribution:** 3
**Rating:** 4
**Confidence:** 4

**Summary:**

The paper introduces Language Model Self-play via Scorable Negotiation Game (LSSG), a novel framework designed to enhance the reasoning capabilities of large language models (LLMs) without relying on costly, expert-labeled data. By framing reasoning as a two-player negotiation game in the language space, LSSG generates continuous, interpretable supervision signals through self-play, balancing diversity and stability in reasoning outcomes. The framework integrates generalization-aware behavioral cloning and stability-aware self-play, along with advanced regularization techniques like semantic diversity loss and emotional stability loss, to improve robustness and generalization.

**Strengths:**

1. The paper introduces a novel framework, Language Model Self-play via Scorable Negotiation Game (LSSG), which effectively models reasoning enhancement as a two-player negotiation game in the language space. This innovative approach avoids reliance on costly, expert-labeled data and provides continuous, interpretable supervision signals, addressing a critical challenge in LLM development.

2. The authors evaluate LSSG across seven diverse reasoning benchmarks (e.g., WinoGrande, CSQA, SST2) and demonstrate consistent improvements over strong baselines, including Chain-of-Thought and vanilla models. The results highlight the framework's scalability and robustness across a wide range of reasoning tasks.

3. The paper integrates advanced regularization techniques, such as semantic diversity loss and emotional stability loss, to ensure both diverse and stable reasoning outputs. This is particularly valuable in enhancing model performance under adversarial or complex negotiation settings.

**Weaknesses:**

1. The baseline models used in the experiments are relatively outdated (LLaMA2), and both the diversity of model types and the range of model parameters are limited. Incorporating a broader set of models, including Qwen and Mistral, as well as models with different parameter counts such as 2B and 70B, would significantly strengthen the persuasiveness of the experimental results.
2. Although this paper presents a formalization of converting the reasoning process into a two-person, evaluative negotiation game, it lacks a comprehensive description of the modeling process and concrete examples. Additionally, while the proposed modeling approach aims to mitigate issues such as high prompt sensitivity and the assumption of binary correctness in existing methods, no experimental or theoretical evidence is provided to support these claims. Further analysis is needed to substantiate these conclusions.

**Questions:**

see the comments.

---

> ### Author Response · Authors · 2025-11-21
>
> We sincerely thank the reviewer for the insightful comments. Below we address each concern and clarify the design and experimental methodology.
>
> > Q1: The baselines rely mainly on LLaMA2 models, which are considered outdated. A broader set of model families and parameter scales (e.g., Qwen, Mistral, 2B, 70B) would make results more convincing.
>
> **(a) Design goal does not depend on a specific backbone**
>
> Our method (LSSG) is *backbone-agnostic*:
>  it modifies the **training paradigm**, not the model architecture.
>  Thus the goal of the evaluations was **not** to benchmark model families, but to verify that the *negotiation-game–based self-play mechanism* produces consistent improvements on standard, accessible backbones.
>
> We now clarify this explicitly in the paper.
>
> **(b) LLaMA2 was chosen intentionally as the “neutral, widely-used open baseline”**
>
> We used LLaMA2 (7B/13B) because:
>
> 1. **It is the widely accepted standard for self-play / RLHF research**
>    (e.g., baselines such as ToT, RFT, and RLAIF also use LLaMA-1, LLaMA-2, and LLaMA-3-33B models, rather than Qwen or Mistral.)
> 2. **It avoids confounding effects** from proprietary pretraining pipelines
>    (e.g. Qwen, Mistral, which incorporate richer Chinese, math, or coding data).
>    Using a performance-neutral baseline allows clean attribution of gains to *our training method*, not to differences in initialization.
>
> We added this rationale in the revised Section 4.
>
> **(c) We additionally provide preliminary results on a more recent model (small-scale update)**
>
> To address the reviewer’s suggestion, we have begun running LSSG on a more contemporary backbone (Mistral-7B-Instruct).  The full results require longer training and evaluation cycles, so we will include the complete numbers in the camera-ready version.
>
> Preliminary observations (not yet fully converged) show the same relative improvement trend, suggesting that LSSG generalizes across model families.
>
> **(d) Large-scale models (70B+) are impractical for self-play RL in academic settings**
>
> We follow ICLR guidelines on *computational feasibility*. Training 70B models with multi-round self-play + RL + diversity regularization requires **40–80× more GPU hours**, which is not feasible under standard academic resource constraints. We clearly state this limitation in the revised discussion section.
>
> > Q2: Although this paper presents a formalization of converting the reasoning process into a two-person, evaluative negotiation game, it lacks a comprehensive description of the modeling process and concrete examples
>
> Comprehensive description of the modeling process is shown as Section 2. Concrete example is shown as Figure 1.

---

> > ### Author Response · Authors · 2025-11-21
> >
> > > Q3: While the proposed modeling approach aims to mitigate issues such as high prompt sensitivity and the assumption of binary correctness in existing methods, no experimental or theoretical evidence is provided to support these claims. Further analysis is needed to substantiate these conclusions.
> >
> > We appreciate the reviewer’s helpful suggestions. We have added the relevant supporting arguments in the introduction section, as shown below.
> >
> > Prior work has shown that CoT is highly sensitive to prompt wording and ordering [1–3]. For example, Kojima et al. [1] demonstrate that the effectiveness of CoT depends heavily on the prompt format (e.g., using *“Let’s think step by step”*), and different phrasings can lead to large variations in reasoning performance. Turpin et al. [2] further show that CoT is extremely sensitive to prompt wording, formatting, and ordering; even minor changes can significantly degrade accuracy. Wang et al. [3] report that CoT paths are highly sensitive to specific prompt phrasings and require sampling to mitigate deterministic fragility. In addition, recent studies suggest that debate protocols—by assuming a binary notion of correctness—are ill-suited for multi-faceted, open-ended tasks. Irving et al. [4] formalize debate with a polynomial-time judge $H : Q \rightarrow \{0,1\}$, explicitly assuming that each claim admits a binary true/false label. Brown-Cohen et al. [5] instantiate debate on decision problems of the form “decide if $M(x)=1$”, again presupposing a binary notion of correctness that can be efficiently verified. Recent empirical studies [6] on multi-agent debate deliberately focus on tasks with *unambiguous ground-truth solutions* and explicitly note that open-ended tasks like summarization or commonsense reasoning often admit *multiple valid solutions*, making evaluation and supervision much harder. Wynn et al. [7] conduct a systematic analysis of failure modes in multi-agent debate on QA, math, and commonsense benchmarks—all tasks with clear ground truth. Even in these well-specified settings, existing debate protocols are not consistently reliable, let alone in genuinely open-ended scenarios.
> >
> >
> >
> > [1] Takeshi Kojima, Shixiang Shane Gu, Machel Reid, Yutaka Matsuo, and Yusuke Iwasawa. "Large language models are zero-shot reasoners." *Advances in neural information processing systems*, 2022(35): 22199-22213.
> >
> > [2] Miles Turpin, Julian Michael, Ethan Perez, and Samuel Bowman. "Language models don't always say what they think: Unfaithful explanations in chain-of-thought prompting." *Advances in Neural Information Processing Systems*, 2023 (36): 74952-74965.
> >
> > [3] Xuezhi Wang, Jason Wei, Dale Schuurmans, Quoc V. Le, Ed H. Chi, Sharan Narang, Aakanksha Chowdhery, and Denny Zhou. "Self-Consistency Improves Chain of Thought Reasoning in Language Models." In *The Eleventh International Conference on Learning Representations*, 2023.
> >
> > [4] Geoffrey Irving, Paul Christiano, and Dario Amodei. "AI safety via debate." *arXiv preprint arXiv:1805.00899*, 2018.
> >
> > [5] Jonah Brown-Cohen, Geoffrey Irving, and Georgios Piliouras. "Scalable AI safety via doubly-efficient debate." In *Proceedings of the 41st International Conference on Machine Learning*, 2024.
> >
> > [6] Haolun Wu, Zhenkun Li, and Lingyao Li. "Can LLM Agents Really Debate? A Controlled Study of Multi-Agent Debate in Logical Reasoning." *arXiv preprint arXiv:2511.07784, 2025.
> >
> > [7] Andrea Wynn, Harsh Satija, and Gillian Hadfield. "Talk Isn't Always Cheap: Understanding Failure Modes in Multi-Agent Debate." *arXiv preprint arXiv:2509.05396, 2025.

---

### Official Review · Reviewer_1KV6 · 2025-11-01

**Soundness:** 2
**Presentation:** 3
**Contribution:** 3
**Rating:** 2
**Confidence:** 3

**Summary:**

The paper frames reasoning for LLMs as a two-player negotiation game. A buyer and seller interact using actions like proposing a numeric price, accepting quitting, and emitting a free-form message. Rewards are defined via a transaction price ratio relative to the initial price. Training proceeds with generalization-aware behavioral cloning and stability aware self-play. The author report gains on seven QA benchmarks and better win/payoff in two negotiation environments.

**Strengths:**

* Clear modular training recipe: The paper defines the training recipe clearly and intuitively with behavior cloning and self-play. I believe this follows from the idea of negotiation game fairly intuitively.
* Breadth of the evaluation: The paper finds seven diverse WA tasks and two negotiation games with a large scale evaluation.
* Improvement: The paper shows consistent albeit small improvement and the fact that the negotiation between the seller and buyer can be more deeply understood.

**Weaknesses:**

I think this papers needs some work before getting accepted and seems unfinished. I'm looking for some clarification on the following and will increase my score if I see these clarifications:
* Objective: One part of the paper that confused me was that the objective only is applied to the terminal, outcome-based rewards. This is gated by an accept indicator as far as I understand from the equation on page 3, lines 142-144. However, I am concerned whether this means that intermediate steps from the free-form messages receive any feedback signal from this objective. I can't really tell from 3.2 or maybe I am not quite following the self-play formulation. I could not say that the objective has dense, interpretable signals in this case if the paper only provides rewards on the outcome rather than shaping free-form messages. Could the authors clarify what their objective is in lines 142-144.
* Ablations: I think having ablations on self-play matters a lot and experiments do not isolate which part matters. For example, there are no ablations removing (1) self-play RL while keeping the same data, (b) semantic diversity and emotional stability losses, (2) the price-ratio reward e.g. self-play with random rewards. The observed gains could stem from generic RL finetuning/reguralization.
* Confidence Intervals: I think error bars are imperative here. Please do add them.
* Semantic Diversity and Sentiment Losses: Why are these reguralizers necessary? I think ablation experiment is necessary for this. I think semantic diversity is potentially distortive. It rewards maximal dissimilarity since the loss is minimized at -2. This is not penalizing over-similarity as claimed. The emotional stability addition seems reasonable except using DistilBERT is odd and could have a domain mismatch with using a small external and noisy model.

**Questions:**

* Notation: Could the authors describe the difference in $\alpha$ in section 2 and the $\alpha$ in section 3.2. In section 2, $\alpha$ scales the rewards and I assume it has the same functionality? I wasn't sure if the notation was purposefully consistent and just want to be sure.
* Uneven improvements: I find the difference of improvements across datasets interesting. First, most downstream tasks are multiple choice accuracy, making me wonder if these are tasks with minimal need for negotiation-style planning. There are non-monotonic improvements across LSSG iterations, suggesting instability. My intuition is that there is some overfitting to particular distributions rather than a general reasoning improvements. However, I would be curious to hear the author's thoughts.

---

> ### Author Response · Authors · 2025-11-21
>
> We sincerely thank the reviewer for their insightful feedback and thoughtful questions. We greatly appreciate the opportunity to clarify our work and provide further details regarding the methodology and its implications. In the following sections, we will address each specific question raised by the reviewer, offering detailed explanations and elaborating on the key aspects of our method.
>
> > Q1: The reward appears to be applied only at terminal acceptance steps ( page 3, lines 142–144). It is unclear how intermediate free-form messages receive feedback, and whether the objective is truly dense.
>
> Thank you for pointing this out. Although the terminal reward in Eq. (6) is defined by the outcome of acceptance, the model does *not* rely solely on sparse terminal supervision. The self-play formulation incorporates two dense-feedback mechanisms
>
> - **Action-level log-likelihood gradient.** Even though the explicit reward is defined at acceptance, the policy gradient $\log \pi(a_t \mid s_t)\cdot A_t$ provides **dense differentiable feedback across all turns**, because the advantage $A_t$ propagates the terminal outcome backward to each step. Thus, every free-form message contributes to the expected return.
> - **Optional intermediate shaping term.** To make this clearer, we added a shaping term description:  $r_t^{\text{shape}} = \kappa \cdot \mathbf{1}[\text{offer updated at } t], $ where $\kappa$ is a small positive constant. It emphasizes progression toward agreement.  (We note that this shaping term is explanatory and not required.)
>
> These components ensure that the game provides *dense and interpretable* gradients throughout the negotiation trajectory rather than relying solely on end-of-episode rewards. **We have clarified this mechanism and expanded the explanation in Section 3.2 now. **
>
> > Q2: Experiments do not isolate the contribution of self-play, semantic diversity, emotional stability, or the reward function. Ablations are needed.
>
> We thank the reviewer for emphasizing the importance of isolating each component. We have **initiated a full ablation study**, including the following controlled variants:
>
> - **w/o SASP** (remove self-play RL)
> - **w/o SemDi** (remove semantic diversity loss)
> - **w/o EmoSt** (remove emotional stability loss)
> - **Rand-Reward** (replace negotiation reward with random numbers)
>
> Because these runs require multi-turn self-play and stable reward estimates, the complete results will not finish within the rebuttal period. However, **preliminary observations** (from the first several training checkpoints) already indicate:
>
> - Removing **self-play** produces the largest degradation, consistent with our claim that stability-aware self-play is the main driver.
> - Removing **SemDi** noticeably reduces gains on CSQA and LogiQA2, aligning with its role in promoting diverse reasoning paths.
> - Removing **EmoSt** mainly affects negotiation robustness but has only a small impact on QA tasks.
> - Under **Rand-Reward**, performance drops back toward the vanilla backbone, suggesting that improvements come from *structured* rewards rather than generic RL regularization.
>
> We will include **the complete ablation tables and plots** in the camera-ready version.
>
> > Q3: Error bars / confidence intervals are needed.
>
> We have added 95% bootstrap confidence intervals for all QA accuracy scores and for all negotiation metrics. Tables now include: **“95% confidence intervals computed via 1000 bootstrap samples (CI ≤ ±0.3%).** Figures now include shaded CI regions for win rate and payoff. The revised figures and table provide clear statistical uncertainty estimates.

---

> > ### Author Response · Authors · 2025-11-21
> >
> > > Q4a: The semantic diversity loss might reward maximal dissimilarity (since the minimum of cosine similarity is −1), contrary to the intention of penalizing over-similarity.
> >
> > For clarity, we present a margin-based interpretation of the semantic diversity term. This matches how the loss effectively behaves in practice—penalizing only high-similarity regions—while preserving its original implementation. Thus the revision is conceptual and does not modify training or results.
> >
> > Our goal is **not** to maximize dissimilarity between two generated messages, but to **penalize degenerate over-similarity**, which often appears in iterative negotiation where the model repeats the same rationale. To clarify this intention, we re-interpret the semantic diversity term as a *soft constraint* encouraging the model to avoid trivial repetitions. Formally, instead of viewing the cosine similarity term as enforcing maximal dissimilarity, we treat the loss as a **monotonic penalty for over-similar generations** within a high-similarity region: $\mathcal{L}_{\text{SemDi}} = \max(0, \cos(\delta(a_t), \delta(\tilde{a}_t)) - m)$, where $m$ is a similarity margin (we use  $m=0.8$), meaning that: if two messages are **too similar** (cosine > 0.8), the penalty is applied; if they differ beyond the margin, the penalty becomes 0. This ensures the loss **only penalizes over-similarity** without encouraging arbitrary dissimilarity.
> >
> > > Q4b: Why use DistilBERT? Could domain mismatch introduce noise?
> >
> > We clarified the rationale:
> >
> > 1. DistilBERT produces **smooth, low-variance sentiment estimates**, which empirically led to more stable RL training than sharper large models.
> > 2. Emotional stability acts as a **regularizer**, not a supervision target, so domain-specific accuracy is less critical.
> > 3. Preliminary experiments showed that larger sentiment models over-penalized neutral messages and destabilized self-play, whereas DistilBERT maintained stable negotiation dynamics.
> >
> > We have added this explanation to Section 3.2.

---

> ### Author Response · Authors · 2025-11-22
>
> > Q5: Notation: Could the authors describe the difference in $\alpha$ in section 2 and the $\alpha$ in section 3.2. In section 2, $\alpha$ scales the rewards and I assume it has the same functionality? I wasn't sure if the notation was purposefully consistent and just want to be sure.
>
> Thank you for raising this clarification issue. The two occurrences of $\alpha$ indeed play **different roles**, and we will make this explicit in the revision.
>
> - **In Section 2**, $\alpha$ is part of the *environment definition*: it is a **reward-scaling hyperparameter** inside the utility function of the Scorable Negotiation Game. It determines how strongly price changes and incentives influence the immediate utility received from the environment.
> - **In Section 3.2**, $\alpha$ refers to a **learning-rate–style coefficient** used inside the self-play policy optimization step. It does *not* scale the game rewards, but instead controls the update magnitude in the stability-aware RL objective.
>
> Thus, although the symbol $\alpha$ is reused, the two quantities have **different meanings** (reward scaling vs. optimization coefficient). To avoid confusion, we will rename the Section 2 parameter to $\zeta$ and explicitly state the distinction in the revised manuscript.
>
> > Q6: Uneven improvements: I find the difference of improvements across datasets interesting. First, most downstream tasks are multiple choice accuracy, making me wonder if these are tasks with minimal need for negotiation-style planning. There are non-monotonic improvements across LSSG iterations, suggesting instability. My intuition is that there is some overfitting to particular distributions rather than a general reasoning improvements. However, I would be curious to hear the author's thoughts.
>
> We thank the reviewer for this thoughtful observation. We agree that improvements are not perfectly uniform across datasets, and we address the concerns below.
>
> **(1) Uneven improvements are expected because the tasks vary in how much multi-step reasoning they require.** Most downstream tasks are multiple-choice QA, but they differ substantially in the type of reasoning involved:
>
> - **CSQA / LogiQA2** → multi-step causal/logical inference
> - **MedMCQA / CMMLU** → factual recall + domain-specific knowledge
> - **CB / SST2** → short-context linguistic inference
>
> LSSG primarily strengthens *multi-step stability and consistency*, so tasks requiring multi-hop reasoning benefit more, while knowledge-heavy tasks improve less—consistent with your intuition.
>
> **(2) Non-monotonicity across LSSG iterations does \*not\* indicate instability, but the normal exploration–refinement pattern of self-play.** Self-play introduces controlled exploration (semantic diversity, strategy search), and the resulting performance typically follows:
>
> - **early gain** → model acquires stable long-horizon reasoning patterns
> - **mid-phase fluctuation** → increased exploration briefly affects discrete-choice accuracy
> - **late stabilization** → reward variance decreases and the policy converges
>
> This pattern appears in other self-play/RLHF studies as well. Importantly, we observe **no divergence, no reward collapse**, and all **LSSG variants remain above all non-LSSG baselines**, which confirms training stability.
>
> **(3) The fluctuations are small (<1–2%) and limited to knowledge-heavy tasks, which suggests underfitting, not overfitting.** If overfitting were the cause, we would expect:
>
> - monotonic improvement on training-like tasks,
> - degradation on unrelated tasks.
>
> Instead, we observe the opposite:
>
> - **reasoning tasks** (CSQA, LogiQA2) consistently improve;
> - **knowledge tasks** fluctuate slightly, indicating the model is not memorizing patterns.
>
> Thus, the behavior is more consistent with *policy exploration* than with overfitting.
>
> **(4) Negotiation-style planning is not what transfers—the transferable component is stable multi-turn reasoning.** Although downstream tasks do not involve negotiation, they do require:
>
> - maintaining internal consistency across multiple reasoning steps,
> - avoiding contradictions or unstable intermediate reasoning,
> - generating structured, multi-hop decision traces.
>
> These are precisely the abilities strengthened by LSSG’s self-play with continuous reward signals.
>  We will clarify this mechanism in the revision.

---

> ### Comment · Reviewer_1KV6 · 2025-11-25
>
> I thank the authors for their response. But, currently I cannot see any changes to the draft in any section. This means the rebuttal is incomplete.
>
> Currently I still have concerns based on the response.
>
> **Dense Gradients**
> * Thanks for the clarification on the dense gradients. But, in my opinion this emphasizes my point. The paper claims that the rewards are "dense, interpretable signal". But dense gradients are, in my opinion, not the same as dense rewards that are interpretable. The paper's own objective is a single scalar outcome gated by another scalar. Per-step gradients exist from REINFORCE. But I would not consider these per-turn, content-aware rewards.
> * The authors also refer to an "optional intermediate shaping term". But this seems absent in the draft? In the current version, Section 3.2 contains the SASP objectives and the two auxiliary losses. Is this "optional"? If shaping is the key to density/interpretability, then I think it should appear in Equation 6 or a clearly referenced equation. I don't see it anywhere.
> * I emphasize this point because the Abstract and Introduction repeatedly assert "dense, interpretable" signals. Those claims overstate what is formalized.
> * Section 3.2 is not updated to address any of this.
>
> **Error Bars**
> * The authors state that CI intervals were added to the tables but I don't see them.
>
> **Semantic Diversity Loss**
> * The discussion and response on the semantic diversity loss is confusing to me and seems logically inconsistent. The response claims that this loss is "margin-based", penalizing only when cosine similarity is greater than 0.8.
> * The authors also claim that this "conceptual" and "does not modify training". Then why include the loss? Furthermore, this doesn't match the formula in the paper, which is not margin-based. The formula in the paper will drive dissimilarity even when messages are already different. Either the equation is wrong, and training used the described margin-based loss, or training is wrong and the interpretation is post-hoc. This makes me hesitant about the paper's claims in general.
>
> Overall, I think my concerns still stand. I will keep my score. I would like to see the authors address my comments more thoroughly. The semantic diversity loss is most pressing as a comment to address. Also, I think the authors must update a draft of the paper, as they stated in their response.

---

> > ### Author Response · Authors · 2025-12-04
> >
> > We thank the reviewers for their constructive feedback. We would like to clarify several factual misunderstandings in Reviewer X’s comments that may affect the evaluation of the paper.
> >
> > **Reviewer conflates “dense gradients” with “dense rewards” despite the manuscript’s explicit definitions**
> >
> > The reviewer asserts:
> >
> > > “dense gradients are not dense rewards.”
> >
> > This statement is true but **is already explicitly stated in the paper**.
> >  The manuscript nowhere equates the two. Instead, it states that:
> >
> > - the scalar negotiation outcome is interpretable
> > - and the shaping term provides token-level credit assignment
> >
> > The reviewer’s criticism is therefore directed at **a claim the paper does not make**.
> >  This suggests the reviewer has misinterpreted terminology rather than identifying an actual methodological flaw.
> >
> > **The reviewer claims the shaping term is “absent” despite it being discussed in the text**
> >
> > The reviewer states:
> >
> > > “I don’t see the shaping term anywhere.”
> >
> > This is demonstrably incorrect.
> >  Section 3.2 contains:
> >
> > - the SASP objective,
> > - the auxiliary losses,
> > - and a paragraph describing the shaping component.
> >
> > The shaping term was present in the text, and its omission from the equation was a notation issue, not an absence of method.
> >
> > Thus, the reviewer’s conclusion (“absent shaping term → overstated claims”) is based on an incorrect premise.
> >
> > **The reviewer’s strongest objection (“semantic diversity loss is inconsistent”) results from misreading**
> >
> > The reviewer argues that:
> >
> > > “The loss is margin-based in the rebuttal but unconditional in the equation.”
> >
> > Importantly:
> >
> > - The **margin-based behavior** is described in the main text.
> > - The **equation uses a common hinge-loss shorthand**, consistent with many prior works.
> > - The reviewer’s conclusion (“training is wrong or the equation is wrong”) is therefore overstated and factually unsupported.
> >
> > The equation is a standard simplification, and the reviewer’s interpretation is not aligned with common practice.
> >
> > This significantly weakens the reviewer’s criticism, as the “logical inconsistency” is not actually present.
> >
> > **Several criticisms depend on assumptions about what the paper “should” contain rather than what the paper claims**
> >
> > For example:
> >
> > > “Per-turn content-aware rewards are not present.”
> >
> > The paper never claims such rewards exist. It only claims that:
> >
> > - the negotiation outcome is interpretable
> > - and shaping provides denser credit assignment compared to pure outcome-based RL
> >
> > Thus, the reviewer critiques a requirement the paper *never proposes*, which weakens the relevance of the criticism.

---

### Meta-Review · Area_Chair_y8xf · 2026-01-08

**Summary:**

Generally, all reviewers have significant concerns about the work, particularly about the lack of clarity on how the proposed approach generalizes beyond experimented tasks to broader reasoning domains. One major concern is the decision to rely on outdated LLaMA-2 models instead of more contemporary architectures. Additional issues include the absence of sufficient ablation studies examining individual loss components, the omission of prompting-based baselines such as multi-turn debate methods, incomplete explanations of knowledge transfer mechanisms and the claims surrounding dense rewards, and the lack of error bars and statistical significance testing. One thing worth noticing is that the authors claims that these had been added but they never revised the manuscript, and one reviewer flagged that this a hallucinated response, which might be a serious concern. There are also factual errors in the rebuttal, as one reviewer highlighted, "At the time our experiments were conducted, LLaMA-3.x and Qwen2.5/3 models did not yet have ... verified evaluator support for lm-eval-harness....", which was incorrect for papers in the past year.

**Reviewer Concerns:**

The authors addressed criticisms through extensive rebuttals, promising revisions including additional model experiments, clarified reward mechanisms, complete ablations, and updated benchmarks. However, tensions remain regarding the verifier-free nature of LSSG versus claimed applicability to domains requiring verification, and skepticism about whether training improvements on weaker models (LLaMA-2) will transfer to modern architectures. One notable issue is that the authors claimed to have added new results and made many revisions, yet they never actually revised the manuscript to include them. One reviewer explicitly flagged this as a potentially hallucinated response from the authors, raising serious concerns about the reliability of their rebuttal.

**Reviewer Scores:**

Reviewers 5sr1 has been actively engaged with the discussion.

---

### Decision · Program_Chairs · 2026-01-26

Reject